# The Relationship Between No-Regret Learning and Online Conformal Prediction

**Ramya Ramalingam** [1]  **Shayan Kiyani** [1]  **Aaron Roth** [1]

## Abstract

Existing algorithms for online conformal prediction—guaranteeing marginal coverage in adversarial settings—are variants of online gradient descent (OGD), but their analyses of worst-case coverage do not follow from the regret guarantee of OGD. What is the relationship between no-regret learning and online conformal prediction? We observe that although standard regret guarantees imply marginal coverage in i.i.d. settings, this connection fails as soon as we either move to adversarial environments or ask for group conditional coverage. On the other hand, we show a tight connection between *threshold calibrated* coverage and swap-regret in adversarial settings, which extends to group-conditional (multi-valid) coverage. We also show that algorithms in the *follow the regularized leader* family of no regret learning algorithms (which includes online gradient descent) can be used to give group-conditional coverage guarantees in adversarial settings for arbitrary grouping functions. Via this connection we analyze and conduct experiments using a multi-group generalization of the ACI algorithm of Gibbs & Candes (2021).

## 1. Introduction

In prediction problems over a label space $\mathcal{Y}$, a popular method for quantifying uncertainty is to produce *prediction sets* $C(x) \subseteq \mathcal{Y}$ that contain subsets of the label space. Given features $x$, the intended semantics of $C(x)$ is that the true label $y$ will fall into the prediction set (i.e. will be *covered* by the prediction set) with some specified probability, say $90\%$. A-priori producing prediction sets is a very high dimensional problem: there are $2^{|\mathcal{Y}|}$ possible prediction sets, which becomes intractable to enumer-

[1]Department of Computer Science, University of Pennsylvania. Correspondence to: Ramya Ramalingam <ramya23@seas.upenn.edu>.

*Proceedings of the $42^{nd}$ International Conference on Machine Learning*, Vancouver, Canada. PMLR 267, 2025. Copyright 2025 by the author(s).

ate over for even moderately large label spaces. But a key insight of the conformal prediction literature (see e.g. (Angelopoulos & Bates, 2021)) is that given an arbitrary *non-conformity score* $f : \mathcal{X} \times \mathcal{Y} \to \mathbb{R}_{\geq 0}$, there is a 1-dimensional family of nested prediction sets defined as $C_\tau(x) = \{y \in \mathcal{Y} : f(x, y) \leq \tau\}$ that we can optimize over, and — simply by adjusting $\tau$, we can obtain *marginal* coverage at any desired rate $q \in (0, 1)$. If there is a data distribution $\mathcal{D}$, marginal coverage at a rate of $q$ corresponds to the guarantee that $\Pr_{(x,y) \sim \mathcal{D}}[y \in C_\tau(x)] = q$. Stronger guarantees of *groupwise* conditional coverage and (threshold-calibrated) "multivalid" coverage have also been recently developed (Jung et al., 2021; Gupta et al., 2022; Bastani et al., 2022; Jung et al., 2023; Noarov et al., 2023; Gibbs et al., 2025), which ask for coverage to hold conditionally on various events. These methods involve learning a threshold *model* $\tau : \mathcal{X} \to \mathbb{R}$, and producing prediction sets of the form $C_\tau(x) = \{y : f(x, y) \leq \tau(x)\}$. Group conditional coverage starts with a collection of groups $g_1, \ldots, g_k$ represented as indicator functions $g_i : \mathcal{X} \to \{0, 1\}$ (which can be arbitrary and intersecting) and asks that $\mathbb{E}_{(x,y) \sim \mathcal{D}}[y \in C_\tau(x)|g_i(x) = 1] = q$ for each group $g_i$. Multivalid coverage asks that the prediction sets simultaneously satisfy group conditional coverage while also being *threshold calibrated* — i.e. it further conditions on the threshold value $\tau(x) = v$ (in a manner similar to calibration), and asks that for all groups $g_i$ and all threshold values $v$: $\mathbb{E}_{(x,y) \sim \mathcal{D}}[y \in C_\tau(x)|g_i(x) = 1, \tau(x) = v] = q$. The algorithm proposed by Jung et al. (2023) for obtaining groupwise coverage over a set of groups learns $\tau(x)$ by minimizing pinball loss over the class of linear combinations of group indicator functions.

It is also possible to obtain coverage guarantees in online adversarial settings in which there is no distribution $\mathcal{D}$, but instead an arbitrary sequence of examples $(x_t, y_t)$ that arrive sequentially. Here we ask for empirical coverage — the threshold (or function) $\tau_t$ can now be updated over time, and marginal coverage over $T$ rounds corresponds to the requirement that $1/T \sum_{t=1}^{T} \mathbf{1}[y_t \in C_{\tau_t}(x_t)] = q$. Groupwise and multivalid coverage can be similarly defined in the sequential setting. Gibbs & Candes (2021) and Gupta et al. (2022) independently studied coverage in online adversarial settings and proposed quite different algorithms. Gibbs & Candes (2021) gave a very lightweight algorithm which

implements online gradient descent on the pinball loss, and prove that it guarantees marginal coverage. This algorithm chooses $\tau_t$ every day independently of the context $x_t$. They give a custom analysis of the coverage properties of their algorithm, however, and do not derive them from the regret guarantees of online gradient descent. Gupta et al. (2022) (later refined by Bastani et al. (2022)) on the other hand give a more complex algorithm modeled on techniques for sequential calibration, and prove that it obtains multivalid coverage with respect to an arbitrary collection of group functions. As it promises groupwise coverage, it necessarily chooses $\tau_t$ as a function of $x_t$. This suggests a number of interesting questions:

1. If a sequence of thresholds $\tau_t$ has no *regret* with respect to the pinball loss, does this on its own guarantee coverage? Are there circumstances in which it does? Is there a stronger form of regret that does?

2. Can algorithms for guaranteeing *groupwise* regret with respect to pinball loss (e.g. (Blum & Lykouris, 2020; Lee et al., 2022)) similarly be used to give groupwise or multivalid coverage guarantees?

3. Alternately, if the guarantees of Gibbs & Candes (2021) do not follow from the regret guarantee of online gradient descent, can we identify a broader class of algorithms that offer these guarantees and generalize them to offer multigroup (rather than just marginal) coverage guarantees?

In this paper we provide answers to these questions.

## 1.1. Our Results

### 1.1.1. REGRET AND COVERAGE

First we consider the relationship between different kinds of regret that a sequence of thresholds $\tau_t$ can have with respect to the pinball loss objective, and how they correspond to coverage guarantees.

**External Regret** A sequence of thresholds $\tau_1, \ldots, \tau_T$ is said to have no *external* regret if their average pinball loss is no larger than that of the best fixed threshold $\tau^*$ chosen in hindsight. This is the kind of regret guarantee offered by algorithms like online gradient descent and multiplicative weights (Arora et al., 2012). We observe that in adversarial settings, a no external regret guarantee on the thresholds $\tau_t$ does *not* guarantee non-trivial coverage (similar observations have previously been made (Gibbs & Candes, 2021)), but show that it does if the algorithm chooses its threshold independently of any context $x_t$ (as ACI does) and outcomes $y_t$ are drawn i.i.d. from an unknown distribution $\mathcal{D}$. We then turn our attention to *groupwise* regret guarantees. A groupwise external regret guarantee promises no

external regret not just marginally over the whole sequence of rounds $\{1, \ldots, T\}$, but simultaneously on each subsequence $S(g_i) = \{t : g_i(x_t) = 1\}$ corresponding to rounds on which the examples are members of group $g$. Algorithms promising no groupwise regret must receive *context* $x_t$ at each round before they make their prediction specifying which groups the current example is a member of. We show that even when the examples $(x_t, y_t)$ are drawn i.i.d. from a distribution $\mathcal{D}$, contextual algorithms obtaining no external regret (and hence any algorithm obtaining no groupwise external regret) do not necessarily obtain any non-trivial coverage bounds — because even in i.i.d. settings, the context can correlate the prediction and the outcome.

**Swap Regret** We then turn our attention to *swap regret*, which corresponds to a guarantee of no external regret conditional on the value of the threshold played — i.e. a no swap regret guarantee corresponds to a guarantee of no external regret simultaneously on each subsequence $S(v) = \{t : \tau_t = v\}$ defined by threshold values $v$. There exist many efficient algorithms for guaranteeing no swap regret for convex losses (Blum & Mansour, 2007; Foster & Vohra, 1999; Dagan et al., 2024; Peng & Rubinstein, 2024). There also exist efficient algorithms for obtaining *group-conditional* swap regret for arbitrary polynomially sized collections of intersecting groups (Lee et al., 2022; Noarov et al., 2023). We show that (under mild smoothness assumptions on the distribution), threshold calibrated coverage is equivalent to swap regret in the sense that any algorithm for guaranteeing no swap regret with respect to the pinball loss produces thresholds that guarantee threshold calibrated coverage at the target rate, and vice versa. This tight connection carries over to group-conditional swap regret — group conditional swap regret is equivalent to multivalid coverage over the same group structure. This connection holds even for algorithms that use context. This gives new algorithms for guaranteeing group conditional multivalid coverage.

### 1.1.2. COVERAGE GUARANTEES BEYOND REGRET

We then turn our attention to generalizations of the "ACI" guarantee that Gibbs & Candes (2021) prove for online gradient descent on the (1 dimensional) pinball loss. Gibbs & Candes (2021) analyze their algorithm by showing that 1) the marginal mis-coverage rate is proportional to the magnitude of the threshold used at the final iterate, and that 2) all iterates (and so in particular the final one) are bounded. We generalize this result in two ways. First, given a collection of groups $\mathcal{G}$, we consider multi-dimensional problems in which we minimize the pinball loss of a function $\tau_t(x) = \langle \theta^t, g(x_t) \rangle$ defined as a $|\mathcal{G}|$-dimensional linear function of the group indicator functions (mirroring the form of $\tau(x)$ used to obtain group conditional coverage in batch settings in (Jung et al., 2023)). We show that if we

optimize the pinball loss of $\tau_t(x)$ using any algorithm from the "follow the regularized leader" (FTRL) family of no-regret algorithms (Shalev-Shwartz et al., 2012) (a family that includes online gradient descent, but also multiplicative weights and many other no regret learning algorithms), then the coverage rate within each group $g_i$ can be bounded as a function of the magnitude of $\theta_i^T$ (the coordinate of the parameter vector corresponding to group $i$) and the gradient of the regularization function used to instantiate FTRL. This generalizes the bound proven in (Gibbs & Candes, 2021) for 1-dimensional online gradient descent (which is an instance of FTRL regularized by the Euclidean norm). We then prove that when using $|\mathcal{G}|$-dimensional online gradient descent for groupwise coverage, it is possible to bound the magnitude of the maximum coordinate of $\theta^T$ by $O(\sqrt{T})$ even when the group functions need not be binary, and can be general weighting functions $g_i : \mathcal{X} \rightarrow [0, 1]$. This implies a $O(\sqrt{T})$ groupwise coverage bound. We show that this is tight (even in 1-dimension) for *real valued* weighting functions by demonstrating an $\Omega(\sqrt{T})$ lower bound — but conjecture a better rate for binary-valued group indicator functions. Finally, we perform an experimental evaluation of this algorithm, and compare it to the online algorithm for guaranteeing multivalid coverage given by (Bastani et al., 2022). We show that our method converges faster to the desired coverage rate. Further, though our upper-bound on the rate of the maximum coordinate of $\theta_T$ grows with $T$, empirically we see in each experiment (using binary groups) that it grows much slower and remains very small over the full transcript. We conjecture (but cannot prove) that for binary groups, the norm of $\theta_T$ can be bounded by a much more slowly growing function of $T$ (or perhaps can be bounded as a function of only $k$, the number of groups).

### 1.2. Related Work

Online conformal prediction was introduced by Gibbs & Candes (2021), who gave the "ACI" (Adaptive Conformal Inference) algorithm, and noted that it was an instantiation of 1-dimensional online gradient descent on the pinball loss — but that the coverage bound did not follow from the standard regret analysis of online gradient descent. This spurred a number of follow up works that modified or refined the original ACI analysis (Gibbs & Candès, 2022; Feldman et al., 2022; Lekeufack et al., 2024; Angelopoulos et al., 2024; Bhatnagar et al., 2023), some of them by making explicit connections to algorithms which guarantee more refined adaptive regret bounds (Gibbs & Candès, 2022; Bhatnagar et al., 2023) — but the worst-case coverage bounds are never derived via the regret bounds, which are used to make auxiliary claims (such as convergence to the true quantile of the loss in stationary or slowly changing environments).

Our generalization of the 1-dimensional ACI bounds to

groupwise coverage bounds was independently and concurrently discovered by Angelopoulos et al. (2025). Angelopoulos et al. (2025) develop their bounds as part of an elegant and general theory of *gradient equilibrium*, whereas we restrict attention to pinball loss and the online coverage problem. We study the more general class of follow the regularized leader algorithms, whereas they restrict attention to online gradient descent.

For some additional related work, please see Appendix B.

## 2. Definitions

Define a joint feature-label space $(\mathcal{X}, \mathcal{Y})$. In uncertainty quantification, one of our goals is to learn prediction sets $C : \mathcal{X} \rightarrow 2^{\mathcal{Y}}$ that satisfy certain probabalistic guarantees. Specifically, for some specified coverage rate $q$, we would like to produce sets that include the true label with probability $q$. Conformal prediction simplifies this problem by defining a collection of nested sets parametrized by a single variable (call it $\tau$) in the following manner:

$$C_\tau(x) = \{y \in \mathcal{Y} : f(x, y) \leq \tau\} \qquad (1)$$

where $f : \mathcal{X} \times \mathcal{Y} \rightarrow \mathbb{R}_{\geq 0}$ can be any arbitrary function, called a *non-conformity measure*. Conformal methods can be viewed as predictors that choose the "correct" values of $\tau$. In distributional settings, to achieve an exact coverage guarantee of the form $\mathbb{P}_{(x,y) \sim \mathcal{D}}[y \in C_\tau(x)] \approx q$, this can be done by using training data to get an estimate $\hat{\tau}$ of the $q$-th quantile of non-conformity score values induced by $\mathcal{D}$.

The $q$-th quantile of a distribution minimizes the expectation of a convex function called the *pinball loss*, defined as:

$$p_q(\hat{\tau}, \tau) = \begin{cases} q(\tau - \hat{\tau}) & \text{if } \tau \geq \hat{\tau} \\ (q - 1)(\tau - \hat{\tau}) & \text{if } \tau < \hat{\tau} \end{cases}$$

The procedure used in conformal prediction can therefore also be seen as finding an estimate $\hat{\tau}$ of the true value $\tau$ that minimizes the expected pinball loss. But in the adversarial setting, there is no longer any distribution over which to estimate a fixed parameter $\hat{\tau}$. Instead, at each round $t$, we may be given features $x_t$ (if we are in the "contextual" setting) and use it to predict a parameter $\hat{\tau}_t$ (and correspondingly the prediction set $C_{\hat{\tau}_t}$). Then we receive the true label $y_t$. In the non-contextual setting we must choose $\hat{\tau}_t$ without any $x_t$, solely based on the history thus far. Note that $y_t \in C_{\hat{\tau}_t}$ iff $\hat{\tau}_t \geq \tau_t$. Thus, we may view online conformal prediction as a sequential prediction task, where over $T$ rounds,

1. The adversary chooses a joint distribution over contexts $x_t \in \mathcal{X}$ and non-conformity score thresholds $\tau_t \in [0, 1]$.

2. The learner, given a realized context $x_t$, makes a prediction $\hat{\tau}_t$ of the score threshold.

3. The learner receives a realized threshold $\tau_t$.

Given a desired coverage level $q$, the goal is to make predictions such that $\frac{1}{T}\sum_{t=1}^{T} \mathbf{1}[\hat{\tau}_t \geq \tau_t] \approx q$. Note that for simplicity, we abstract away the true label $y_t$ and non-conformity score $f_t$ — implicitly, $\tau_t = f_t(x_t, y_t)$. We assume that all thresholds $\tau_t$ are bounded in $[0, 1]$. In practice, non-conformity measures can be normalized to ensure this holds.

**Definition 2.1** (Transcript). A *transcript* $\Pi_T = \{(x_t, \tau_t, \hat{\tau}_t)\}_{t=1}^{T}$ denotes a sequence of contexts, outcomes and predictions in the sequential prediction setting. Let $\Pi^* = (\mathcal{X} \times [0, 1] \times [0, 1])^*$ denote the set of all transcripts.

### 2.1. Coverage

**Definition 2.2** (Coverage, Coverage Error). Given a transcript $\Pi_T = \{(x_t, \tau_t, \hat{\tau}_t)\}_{t=1}^{T}$, the *coverage* of $\Pi_T$ is defined as:

$$\text{Cov}(\Pi_T) = \frac{1}{T}\sum_{t=1}^{T} \mathbf{1}[\hat{\tau}_t \geq \tau_t]$$

For a desired coverage rate $q \in (0, 1)$, we have *coverage error* $\gamma$ with respect to $q$ if $|\text{Cov}(\Pi_T) - q| \leq \gamma$.

One can examine coverage not just marginally over the transcript, but also over subsequences of the transcript that define groups within the full sequence. These groups may be defined by context, the predicted threshold, or even the past transcript, as long as membership can be determined at the start of the round, as the learner makes a prediction.

**Definition 2.3** (Group). A group $G : \Pi^* \times \mathcal{X} \times [0, 1] \to [0, 1]$ is a mapping from a transcript, context, and threshold to a real value indicating group-membership.

If $G(\Pi_t, x, \tau) \in \{0, 1\}$ for all $\Pi_t \in \Pi^*, x \in \mathcal{X}, \tau \in [0, 1]$, we call $G$ a *binary group*. If $G(\Pi_t, x, \tau_1) = G(\Pi_t, x, \tau_2)$ for all $\Pi_t \in \Pi^*$, $x \in \mathcal{X}$ and $\tau_1, \tau_2 \in [0, 1]$, we call $G$ a *prediction-independent group*.

We allow group-membership to be real-valued to be able to model scenarios involving partial or probabalistic membership in a group, but in most cases group-membership is deterministic, and only binary groups need be used. The value of a prediction-independent group cannot depend on the prediction being made on that day - in Section 4, our results for group conditional coverage hold only for such groups.

**Definition 2.4** (Groupwise Coverage, Group Size). Given a transcript $\Pi_T = \{(x_t, \tau_t, \hat{\tau}_t)\}_{t=1}^{T}$ and a set of groups $\mathcal{G}$, the *coverage* of group $G \in \mathcal{G}$ over $\Pi_T$ is defined as:

$$\text{Cov}(\Pi_T, G) = \frac{1}{T_G}\sum_{t=1}^{T} \mathbf{1}[\hat{\tau}_t \geq \tau_t] \cdot G(\Pi_t, x_t, \hat{\tau}_t)$$

where we define the *size* of the group $T_G = \sum_{t=1}^{T} G(\Pi_t, x_t, \hat{\tau}_t)$. For a desired coverage rate $q \in (0, 1)$, we have *groupwise coverage error* $\gamma$ with respect to $q$ if $|\text{Cov}(\Pi_T, G) - q| \leq \gamma$ for all $G \in \mathcal{G}$.

Since our setting reduces the problem of building prediction sets to one of predicting a sequence of real-valued parameters, we may ask, in addition to achieving a coverage guarantee, that the sequence of predictions satisfies coverage over groups defined by the level sets of the predicted threshold value.

**Definition 2.5** (Threshold-calibrated coverage). Given a transcript $\Pi_T$ and a desired coverage rate $q \in (0, 1)$, we have *threshold-calibrated coverage* with coverage error $\gamma$, if we have groupwise coverage error $\gamma$ with respect to the collection of groups $\mathcal{G} = \{G_\tau : \forall \tau \in [0, 1]\}$, where $G_\tau$ is a binary group including all time-steps $t$ for which $\hat{\tau}_t = \tau$.

**Definition 2.6** (Multivalid coverage). Given a transcript $\Pi_T$, a set of groups $\mathcal{G}$, and a desired coverage rate $q \in (0, 1)$, we have *multivalid coverage* with coverage error $\gamma$, if we have groupwise coverage error $\gamma$ with respect to the new collection of groups $\mathcal{H} = \{H_{G,\tau} : \forall \tau \in [0, 1], G \in \mathcal{G}\}$, where $H_{G,\tau}(\Pi_t, x_t, \hat{\tau}_t) = G(\Pi_t, x_t, \hat{\tau}_t) \cdot \mathbf{1}[\hat{\tau}_t = \tau]$ for all $G \in \mathcal{G}, \tau \in [0, 1]$.

### 2.2. Regret

**Definition 2.7** (Φ-regret (Greenwald & Jafari, 2003)). Given a transcript $\Pi_T = \{(x_t, \tau_t, \hat{\tau}_t)\}_{t=1}^{T}$, an allowable action space of predictions $\mathcal{A}$, and a loss function $l : \mathcal{A} \times \mathcal{A} \to \mathbb{R}$, the *regret* with respect to the loss function $l$ with respect to a strategy modification rule $\phi : \mathcal{A} \to \mathcal{A}$ is:

$$r(\Pi_T, l, \phi) = \sum_{t=1}^{T} l(\hat{\tau}_t, \tau_t) - l(\phi(\hat{\tau}_t), \tau_t)$$

For any collection of strategy modification rules $\Phi$, we say that $\Pi_T$ has Φ-regret $\gamma$ with respect to $l$ if $r(\Pi_T, l, \phi) \leq \gamma$ for all $\phi \in \Phi$.

A transcript has *external regret* if it has Φ-regret with respect to the set of all constant strategy modification rules (of the form $\phi(x) = y$ for all $x \in \mathbb{R}$), and it has *swap regret* if it has Φ-regret with respect to the set of all strategy modification rules. Existing swap-regret algorithms such as (Blum & Mansour, 2007) achieve regret guarantees that have a dependence on the size of the action set $\mathcal{A}$. Therefore, though the true parameter values $\{\tau_t\}_{t=1}^{T}$ are allowed to take any value in $[0, 1]$, we will consider a discretized prediction space parametrized by parameter $n$, i.e. predicted values $\hat{\tau}$ can take values only in the set $\mathcal{A}_n = \{0, 1/n, 2/n, \cdots, 1\}$, and the set of strategy modification rules being compared to is $\Phi_n$, the collection of all strategy modification rules $\phi : \mathcal{A}_n \to \mathcal{A}_n$. We can similarly define groupwise external and swap regret given some collection of groups $\mathcal{G}$:

**Definition 2.8** (Φ-groupwise regret). Given a transcript $\Pi_T$, an allowable action space of predictions $\mathcal{A}$, a loss function $l : \mathcal{A} \times \mathcal{A} \to \mathbb{R}$, and a set of groups $\mathcal{G}$, the *regret* with respect to the loss function $l$ and group $G$, with respect to a strategy modification rule $\phi : \mathcal{A} \to \mathcal{A}$ is:

$$r(\Pi_T, l, \phi, G) = \sum_{t=1}^{T} \left( l(\hat{\tau}_t, \tau_t) - l(\phi(\hat{\tau}_t), \tau_t) \right) \cdot G(\Pi_t, x_t, \hat{\tau}_t)$$

For any collection of strategy modification rules $\Phi$, we say that $\Pi_T$ has Φ-groupwise regret $\gamma$ with respect to $l$ if $r(\Pi_T, l, \phi, G) \leq \gamma$ for all $\phi \in \Phi$ and $G \in \mathcal{G}$.

*Groupwise external regret* corresponds to Φ-regret with respect to the set of all constant strategy modification rules, and *groupwise swap regret* corresponds to Φ-regret with respect to the set of all strategy modification rules. There exist efficient algorithms for obtaining diminishing groupwise external and swap regret for any polynomial action space and collection of groups $\mathcal{G}$ (Blum & Lykouris, 2020; Lee et al., 2022; Acharya et al., 2024; Deng et al., 2024).

To move between no regret and coverage guarantees, note that it is necessary for the threshold parameters to not be too closely clustered together. Suppose for example we had an empirical distribution defined by $\{\tau_t\}_{t=1}^{T}$ that put all probability mass on a single value $a$. Then the fixed prediction in $\mathcal{A}_n$ closest to $a$ would achieve no swap-regret, but would correspond to coverage over either all rounds or no rounds, thus being bounded away from the desired coverage rate $q$ for any $q \in (0, 1)$. To avoid this kind of scenario, we introduce a smoothness condition that guarantees the parameters we are trying to predict are sufficiently distributed across the support of our probability space.

**Definition 2.9** ($(\alpha, \rho, r)$-smoothness). A distribution $\mathcal{D} \in \Delta[0,1]$ is said to be $(\alpha, \rho, r)$-smooth if for every pair of values $p, q$ such that $0 \leq p \leq q \leq 1$ and $|p - q| \leq 1/r$, we have $\mathbb{P}_{\tau \sim \mathcal{D}}[\tau \in [p, q]] \leq \rho$, and if $|p - q| >= 1/r$, then $\mathbb{P}_{\tau \sim \mathcal{D}}[\tau \in [p, q]] \geq \alpha$.

## 3. Coverage Guarantees Through Regret

In stochastic settings without context, external regret (under some mild smoothness conditions) is sufficient to obtain marginal coverage. To prove this, we first draw a connection between the expected difference in pinball loss between two thresholds and their absolute difference, using a slightly modified version of Proposition 5 from (Gibbs & Candès, 2022):

**Lemma 3.1.** *Fix a distribution $\mathcal{D}$, and let $\tau^*$ be the $q$-th quantile of $\mathcal{D}$. Then, assuming $\mathcal{D}$ is an $(\alpha, \rho, r)$-smooth distribution, for any other threshold $\tau'$,*

$$\frac{\alpha r \cdot (\tau^* - \tau')^2}{2} \leq \mathbb{E}_{\tau \sim \mathcal{D}}[p_q(\tau', \tau) - p_q(\tau^*, \tau)]$$

With this, we can move from a regret bound to a bound on our miscoverage rate. We give a sketch of the proof here, with the full version in the appendix.

**Theorem 3.2.** *Fix a transcript $\Pi_T = \{(\tau_t, \hat{\tau}_t)\}_{t=1}^{T}$ in a setting without context (i.e. in which there are no observable features $x_t$) and where the sequence of labels is drawn i.i.d. from a fixed distribution, i.e. $\tau_t \sim \mathcal{D}$ for all $t \in [T]$. If $\mathcal{D}$ is $(\alpha, \rho, r)$-smooth, and if $\Pi_T$ has external regret $\gamma$ with respect to the pinball loss $p_q$, then the set of predicted thresholds has marginal coverage error:*

$$|\text{Cov}(\Pi_T) - q| \leq \sqrt{\frac{2\rho(\gamma + 2\epsilon)}{T\alpha}} + \frac{\epsilon}{T}$$

*with probability at least $1 - 6\exp\left(-\frac{\epsilon^2}{2T}\right)$.*

*Proof Sketch.* We are given an upper-bound on the realized regret with respect to $p_q$, which with high probability is close to the expected regret (using Azuma's inequality). We then bound the sum of squared differences between the optimal threshold $\tau^*$ (which minimizes simultaneously the expected pinball loss and deviation of expected coverage from $q$) and the predicted thresholds $\hat{\tau}_t$ using Lemma 3.1. The smoothness condition implies thresholds that are close together must have similar expected coverages, and another application of Azuma's inequality proves that this is (with high probability) close to the realized coverage. $\square$

When we have sublinear external regret, the bound above goes to zero as $T$ increases. But in adversarial settings, there is no such connection between external regret and coverage even in the non-contextual setting.

**Example 3.3.** *Define the transcript $\Pi_T = \{(\tau_t, \hat{\tau}_t)\}_{t=1}^{T}$ in the non-contextual setting where the predicted threshold $\hat{\tau}_t = 0.4$ for odd $t$ and $0.9$ for even $t$, and the adversary chooses $\tau_t = 0.5$ for odd $t$ and $\tau_t = 1$ for even $t$. On each day $t$, the loss with respect to $p_q$ (for $q = 0.5$) is $0.1$, and since the true thresholds distribute evenly over the set $\{0.5, 1\}$, the best fixed threshold $\tau^*$ in hindsight is the median $0.75$ which achieves a loss of $0.25$ every day. Therefore $\sum_{t=1}^{T} p_q(\hat{\tau}_t, \tau_t) - p_q(\tau^*, \tau_t) \leq 0$, i.e this transcript has no regret with respect to pinball loss at the level $q = 0.5$. However, the predicted threshold is always lower than the true threshold, and so $\text{Cov}(\Pi_T) = 0$.*

In fact, the connection between low regret on a sequence and achieving low miscoverage on that same sequence falls apart even in i.i.d. settings when we move to groupwise coverage. Theorem 3.2 is driven by the fact that in non-contextual settings, the prediction made each day is independent of the realized outcome drawn from $\mathcal{D}$. When we introduce groups, we move to the contextual setting. As soon as we allow the threshold to depend on context, we find that external regret

no longer implies coverage even in i.i.d. settings because of the correlation that the context introduces between our predictions and the outcomes.

**Example 3.4.** *Define the context space $\mathcal{X} = \{A, B\}$, and suppose we are interested only in marginal coverage over the group $G$ containing all days. The distribution $\mathcal{D}$ over $(x, y)$ pairs is defined such that we randomize uniformly over contexts (i.e. $\mathbb{P}(x = A) = \mathbb{P}(x = B) = 0.5$), and non-conformity score function $f$ is such that $f(A, .) = 0.5$, and $f(B, .) = 1$. Then an algorithm $\mathcal{A}$ that always predicts a threshold $\hat{\tau}_t = 0.4$ when $x_t = A$, and a threshold $\hat{\tau}_t = 0.9$ when $x_t = B$, simulates the environment described in Example 3.3. Thus $\mathcal{A}$ will achieve an expected regret of 0 (which will be arbitrarily close to the realized regret with high probability for sufficiently large $T$), but always a realized coverage of zero.*

To make further connections between regret and coverage, we will need to move to stronger guarantees. Low regret on subsequences for which a prediction is fixed (which are by definition disjoint) allows us to compare the performance of the optimal quantile on that subsequence with the fixed prediction. Thus, given the empirical distribution over these subsequences are smooth, we can draw an equivalence between threshold-calibrated coverage and swap-regret.

**Theorem 3.5.** *Fix a transcript $\Pi_T = \{(x_t, \tau_t, \hat{\tau}_t)\}_{t=1}^T$. If $\Pi_T$ has swap regret $\gamma$ with respect to the pinball loss $p_q$, and the empirical distribution $\mathcal{D}_\tau$ defined by the set $\{\tau_t\}_{t:\hat{\tau}=\tau}$ is $(\alpha, \rho, r)$-smooth for each $\tau \in \mathcal{A}_n$, then the set of predicted thresholds satisfies threshold-calibrated coverage at the level $q$:*

$$|Cov(\Pi_T, G_\tau) - q| \leq \frac{\rho}{2} + \frac{\rho r}{n} + \sqrt{\frac{2\gamma}{T_\tau \alpha r} + \frac{\rho}{\alpha}\left(\frac{1}{r} + \frac{2}{n}\right)}$$

*Proof Sketch.* The swap regret guarantee gives an upper-bound on the regret of the subsequence defined by all time-steps making a fixed prediction $\tau$. Due to convexity of the pinball loss function, the true minimizer of the sum of pinball losses must be close to the minimizer in the discrete set $\mathcal{A}_n$, which in turn can be bound closely to $\tau$ using Lemma A.1 and the regret bound. The smoothness condition implies not a lot of probability weight can be placed in the interval $|M(\tau) - \tau|$, and so the difference in coverage is also small. Since $M(\tau)$ should achieve the desired coverage rate $q$, this gives a bound on the miscoverage on the subsequence defined by any fixed prediction $\tau$ (for all $\tau \in \mathcal{A}_n$). $\square$

The full proof can be found in the appendix. Note that if $\gamma$ (as a function of $T$) grows sublinearly, then the final term in the above inequality vanishes as $T$ becomes arbitrarily large. Several existing swap-regret algorithms (Blum & Mansour,

2007) achieve such rates. We can also move from threshold-calibrated coverage to regret bounds. The proof is similar in idea to Theorem 3.6, so we relegate it to the appendix.

**Theorem 3.6.** *Fix a transcript $\Pi_T = \{(x_t, \tau_t, \hat{\tau}_t)\}_{t=1}^T$. If $\Pi_T$ has threshold-calibrated coverage with coverage error $\gamma$ (at desired coverage rate $q$), and $\mathcal{D}_\tau$ defined by $\{\tau_t\}_{t:\hat{\tau}=\tau}$ is $(\alpha, \rho, r)$-smooth for each $\tau \in \mathcal{A}_n$, then the transcript also has swap regret with respect to the loss $p_q$, such that:*

$$r(\Pi_t, p_q, \phi) \leq \frac{T\gamma^2 \rho}{\alpha^2 r}$$

*for each $\phi \in \Phi$, the collection of all strategy modification rules for action set $\mathcal{A}_n$.*

and so if $\frac{1}{\gamma^2}$ grows at a rate faster than $T$, we achieve sublinear regret. Applying the same analysis in the context of groupwise swap regret (analyzing subsequences determined by a fixed predicted threshold and group inclusion) gives us an analogous relationship between groupwise swap regret and multivalid guarantees.

**Theorem 3.7.** *Fix a transcript $\Pi_T$, and a set of binary groups $\mathcal{G}$. If $\Pi_T$ has groupwise swap regret $\gamma$ with respect to the pinball loss $p_q$, and the empirical distributions $\mathcal{D}_{G,\tau}$ defined by the set $\{\tau_t\}_{t:\hat{\tau}=\tau, t \in G}$ are $(\alpha, \rho, r)$-smooth, then the set of predicted thresholds satisfies multivalid coverage at the level $q$ such that*

$$|Cov(\Pi_T, H_{G,\tau}) - q| \leq \frac{\rho}{2} + \frac{\rho r}{n} + \sqrt{\frac{2\gamma}{T_{G,\tau} \alpha r} + \frac{\rho}{\alpha}\left(\frac{1}{r} + \frac{2}{n}\right)}$$

*for each group $H_{G,\tau}$, defined as $H_{G,\tau}(\Pi_t, x_t, \hat{\tau}_t) = G(\Pi_t, x_t, \hat{\tau}_t) \cdot \mathbf{1}[\hat{\tau}_t = \tau]$.*

**Theorem 3.8.** *Fix a transcript $\Pi_T$, and a set of binary groups $\mathcal{G}$. If $\Pi_T$ has multivalid coverage with coverage error $\gamma$ (at desired coverage rate $q$), and $\mathcal{D}_{G,\tau}$ is $(\alpha, \rho, r)$-smooth for each $\tau \in \mathcal{A}_n, G \in \mathcal{G}$, then the transcript also has groupwise swap regret with respect to the loss $p_q$, such that $r(\Pi_t, p_q, \phi, G) \leq \frac{T_G \gamma^2 \rho}{\alpha^2 r}$ for each $\phi \in \Phi$, the collection of all strategy modification rules for action set $\mathcal{A}_n$, where $T_G$ is the size of group $G$.*

---

**Algorithm 1** Follow The Regularized Leader (pinball loss)

**Input:** Timesteps $T$, regularizer $R : [0, 1] \to \mathbb{R}$, loss parameter $q$
**for** $t = 1, 2, \cdots T$ **do**
    Receive $g_t$ from adversary
    Choose $\theta_t = \arg\min_{\theta \in \mathbb{R}^d} \sum_{s=1}^{t-1} l_t(\theta, \tau_s) + R(\theta)$.
    Predict $\hat{\tau}_t = \langle \theta_t, g_t \rangle$
    Receive $\tau_t$ from the adversary.
    Define loss $l_t(\theta, \tau_t) = \langle \theta, \nabla_\theta p_q(\langle \theta_t, g_t \rangle, \tau_t) \rangle$
**end for**

---

# 4. Coverage Guarantees for FTRL Algorithms

Having established that, in general, external regret guarantees with respect to the pinball loss do not imply non-trivial coverage on their own, either in adversarial settings, or in settings with context (as relevant to groupwise coverage) even in the i.i.d. setting, we turn our attention to a particular (but broad) class of no regret learning algorithms — those in the "follow the regularized leader" (FTRL) family. This class of algorithms includes multiplicative weights, online gradient descent, and many other algorithms. At a high level, an algorithm in the FTRL family receives a loss function $\ell(x, y)$ at every iteration, parameterized by a choice of action $y$ by the adversary and a choice of action $x \in \mathbb{R}^d$ by the learner. The loss is assumed to be linear in $x$ for all choices $y$ of the adversary. An instantiation of FTRL is given by a convex *regularization* function $R : \mathbb{R}^d \to \mathbb{R}$, and the action that FTRL plays at every iteration is $x_t = \arg\min_x \sum_{s=1}^{t-1} \ell(x, y_s) + R(x)$ — the *regularized* empirical risk minimizer on the empirical loss distribution so far. Follow the regularized leader can also be used with loss functions $\hat{\ell}(x, y)$ that are *convex* in $x$. In this case, the algorithm takes as input the linear loss function $\ell(x, y) \doteq \langle x, \nabla_x \hat{\ell}(x_t, y) \rangle$ — defined by the *gradient* of the loss function evaluated at the point $x_t$ the learner plays at round $t$. This reduces to the linear case and obtains the same regret bound (see e.g. (Shalev-Shwartz et al., 2012)). Online gradient descent is an instance of FTRL regularized by the Euclidean norm; multiplicative weights is an instance of FTRL regularized by entropy; other algorithms follow from different regularization functions.

In this section we study coverage guarantees for algorithms in the FTRL family when actions for the learner are parameter vectors $\theta_t \in \mathbb{R}^k$, actions for the adversary are nonconformity scores $\tau_t \in [0, 1]$ and the loss function is $p_q(\langle \theta_t, g_t \rangle, \tau_t)$ — the pinball loss (at a target quantile $q$) of the prediction $\hat{\tau}_t \doteq \langle \theta_t, g_t \rangle$ with respect to $\tau_t$. Here $g_t \in [0, 1]^k$ is the vector of group membership for the example at round $t$, i.e. given $k$ prediction-independent groups $\{G_1, \cdots, G_k\}$, $g_{t,i} = G_i(\Pi_t, x_t, .)$. We show that in this setting, for all algorithms in the FTRL family, the miscoverage rate can be bounded as a function of the magnitude of the parameter $\theta_t$ and the gradient of the regularization function $R(\cdot)$.

**Theorem 4.1.** *For the parametrization of FTRL given in Algorithm 1 with regularization function $R : \mathbb{R}^d \to \mathbb{R}$, for any target coverage rate $q$ and any $T$ the resulting transcript $\Pi_T$ is guaranteed to satisfy groupwise coverage for groups $G_i$ ($i \in [k]$) at the rate:*

$$|Cov(\Pi_T, G_i) - q| \leq \frac{||\nabla R(\theta_{T+1})||_\infty}{T_i}$$

The proof is given in the appendix. This theorem tells

us whenever we can upper-bound $||\nabla R(\theta_{T+1})||_\infty$ by any function that grows sublinearly with $T$, we get a non-trivial groupwise coverage bound. In the following section we do this for online gradient descent, an especially simple instantiation of FTRL. Note that both in this section and Section 5, the analysis for coverage does *not* go through regret — hence no smoothness assumptions need be made on the empirical data distributions.

# 5. Group Conditional ACI

Algorithms such as ACI ("Adaptive Conformal Inference") (Gibbs & Candes, 2021) can be seen as special cases of the connection between FTRL and coverage guarantees we have shown — in particular the special case in which we ask only for marginal coverage, and use gradient descent with step size $\eta$, which is an instantiation of FTRL in which the regularization function $R(\theta) = \frac{1}{2\eta}||\theta||^2$. We give the "gradient descent" implementation of our algorithm in Algorithm 2.

---

**Algorithm 2** Group Conditional ACI (GCACI)

**Input:** Timesteps $T$, number of groups $k$, coverage target $q$, step-size $\eta$
Choose $\theta_1 = \mathbf{0}$.
**for** $t = 1, 2, \cdots T$ **do**
    Receive $\mathbf{g}_t$ from the adversary.
    Predict $\hat{\tau}_t = \langle \theta_t, \mathbf{g}_t \rangle$.
    Receive $\tau_t$ from adversary.
    **if** $\langle \theta_t, \mathbf{g}_t \rangle < \tau_t$ **then**
        $\theta_{t+1} = \theta_t + \eta \cdot q \cdot \mathbf{g}_t$          (Update A)
    **else**
        $\theta_{t+1} = \theta_t - \eta \cdot (1 - q) \cdot \mathbf{g}_t$     (Update B)
    **end if**
**end for**

---

We can instantiate our Theorem 4.1 to bound the groupwise miscoverage of Algorithm 2, as it is a special case of FTRL.

**Lemma 5.1.** *Running Algorithm 2 for any number of rounds $T$ with a coverage target of $q$ for any set of $k$ group functions, we achieve groupwise miscoverage bounded by the following function of $\theta_{T+1}$:*

$$|Cov(\Pi_T, G_i) - q| \leq \frac{||\theta_{T+1}||_\infty}{T_i \eta}$$

If we are able to upper-bound the magnitude of the last iterate of gradient descent as a sublinear function of $T$, we can bound the deviation from desired coverage not just marginally, but groupwise for arbitrary intersecting groups:

**Lemma 5.2.** *When Algorithm 2 is run with step-size $\eta \in (0, 1]$, for any collection of $k$ group functions, any coverage target $q \in (0, 1)$, and every $T$, the iterate $\theta_{T+1}$ has norm bounded as:*

$$||\theta_{T+1}||_\infty \leq \mathcal{O}(\sqrt{\eta T(\eta k + 1)})$$

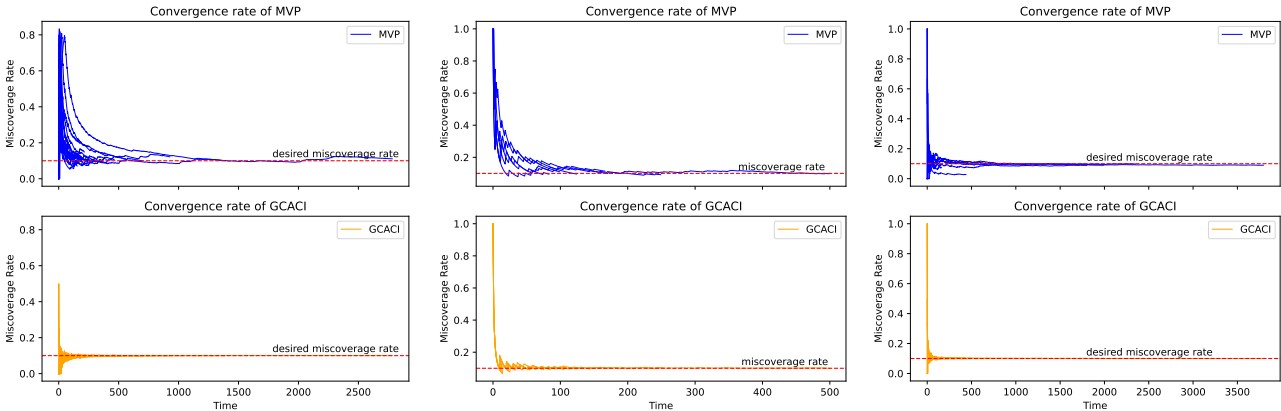

Figure 1: Comparison of convergence rates between GroupACI (GCACI) and MVP for group coverage. Each curve captures the averaged miscoverage over time of a single group. From left to right, the graphs show results on the time series data, the UCI Airfoil Data, and the Folktables data respectively. Note that group size varies within each graph.

Putting these two lemmas together gives us a groupwise coverage bound for Algorithm 2 (GCACI):

**Theorem 5.3.** *Fix any collection of $k$ group functions taking values in $[0, 1]$ and any target coverage rate $q \in (0, 1)$. If we run Algorithm 2 for $T$ rounds with step size $\eta \in (0, 1]$, we achieve groupwise miscoverage bounded by:*

$$|Cov(\Pi_T, i) - q| \le O\left(\frac{\sqrt{\eta T(\eta k + 1)}}{T_i \eta}\right)$$

When we set $\eta = 1$, this gives us a $O(\sqrt{Tk}/T_i)$ groupwise coverage error bound. This analysis is tight even for $k = 1$ if we allow the groups to be real valued.

**Theorem 5.4.** *Let $k = \eta = 1$ and pick any coverage target $q \in (0, 1)$. The sequence of 1-dimensional weighting functions $g_t = \frac{1}{2\sqrt{t-1}}$ together with thresholds $\tau_t = 1$ causes Algorithm 2 to produce parameter vector $\theta_{T+1} \in \Omega(\sqrt{T})$.*

We remark that this lower bound construction seems to require real valued group functions. We conjecture that a much better upper bound on $||\theta_{T+1}||_\infty$ is true for *binary* valued group functions — growing much more slowly with (or perhaps even independently of) $T$. Our experiments support this conjecture, but we are unable to prove it.

## 6. Experiments

In this section we compare the performance of Algorithm 2 with that of the MVP ("multi valid predictor") algorithm (Bastani et al., 2022), that to our knowledge is the only other method for obtaining non-trivial group-conditional coverage guarantees in sequential adversarial settings. We run experiments on the same collection of datasets used to

evaluate MVP in (Bastani et al., 2022). We compare rates of convergence to the desired coverage over all groups. Since the guarantees for our algorithm are more fine grained, and are proven in terms $||\theta_t||_\infty$, we plot also the $L_\infty$ norm of the parameters $\theta_t$ maintained by Algorithm 2 over time. To achieve our derived $O(\sqrt{Tk}/T_i)$ bounds we set the learning rate $\eta = 1$ for these experiments. We also then empirically investigate the relationship between the rate of convergence to the target coverage rate and the learning rate, by measuring the time-step[1] at which the empirical group conditional coverage for the rest of the sequence falls within $\epsilon$ of the desired coverage rate, as a function of $\eta$. We set $\epsilon = 0.01$ for all tests. More fine-grained details on how these experiments were run are given in Appendix C.

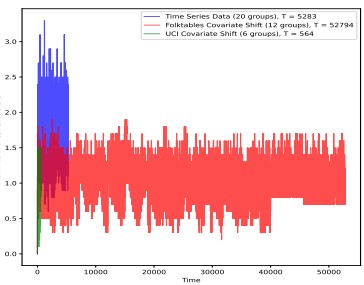

Figure 2: $||\theta_t||_\infty$ over time for all three experiments, when running GCACI.

**Time Series Data** We run both algorithms on stock market data from the WSJ daily price data, which was used to test ACI algorithm's (Gibbs & Candes, 2021) performance for

---

[1]Here, time-step is defined as within the subsequence defined by a group, not the full sequence.

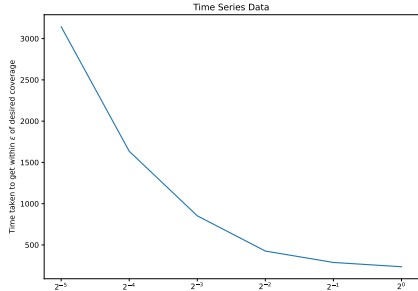 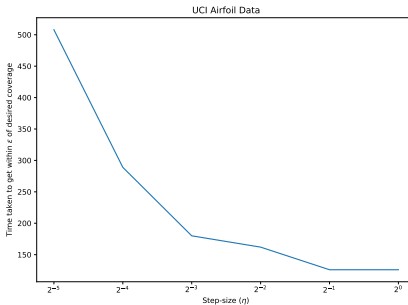 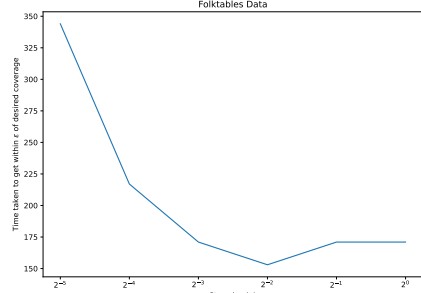

Figure 3: Convergence rate of GCACI as a function of the learning rate $\eta$. Each plot measures (across different chosen learning rates) the earliest time-step at which coverage for each group is within $\epsilon = 0.01$ of the desired coverage for the remainder of the transcript.

marginal coverage. As in (Bastani et al., 2022), we define 20 groups, where group $i$ includes data from time-step $t$ iff $t \equiv 0 \pmod{i}$, and introduce artificial noise to the data to get variability in scores based on group membership.

**Synthetic Distribution Shift (UCI Airfoil Data)** We run both MVP and GCACI on the airfoil dataset from the UCI Machine Learning Repository (Dua & Graff, 2017). In (Bastani et al., 2022), only marginal coverage was tested. Here, we define groups where membership is again defined by time-step, as in the previous section. Since the test set is much smaller ($N = 564$), we define only 6 such groups.

**Natural Distribution Shift (Folktables)** Finally, we compare performance on a covariate shift problem using 2018 Census data from the Folktables repository (Ding et al., 2021). The data ($N = 52794$) is drawn from two different states (CA & PA) to simulate an unknown distribution shift; we use a non-conformity score defined using a quantile regression model trained on a separate part of the CA data. We compare performance across all of the nine race groups in the dataset, groups for both sexes, as well as the full group.

### 6.1. Results

Figure 1 compares how quickly the two algorithms are able to achieve the desired miscoverage rate. We see that convergence is substantially faster for our algorithm — despite the fact that both algorithms have similar $O(\sqrt{T})$ guarantees for worst-case coverage rates. MVP doesn't even converge fully for some smaller-sized groups. We also find that for GCACI, the $O(\sqrt{T})$ upper bound on $||\theta_T||_\infty$ appears to be very loose, at least in the setting of our evaluation. Figure 2 shows that for each experiment, it remains bounded by a small constant, explaining our superior observed coverage performance — because in our experiments, $||\theta_t||_\infty$ remains bounded by a small constant at all iterates $t$, we actually get groupwise coverage rates at $O(1/T)$. This supports our conjecture that much better bounds might be possible for binary group structure. Figure 3 plots how quickly GCACI

converges as a function of the learning rate. We see that as expected, larger learning rates give faster convergence, with the algorithm generally converging most quickly with a learning rate of $\eta = 1$. This naturally trades off with the regret guarantees of follow the regularized leader, which are optimized in the worst case when $\eta = 1/\sqrt{T}$ and vacuous for constant $\eta$.

## Impact Statement

This paper presents work whose goal is to advance the field of Machine Learning. There are many potential societal consequences of our work, none which we feel must be specifically highlighted here.

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

## A. Proofs

**Lemma 3.1.** *Fix a distribution $\mathcal{D}$, and let $\tau^*$ be the $q$-th quantile of $\mathcal{D}$. Then, assuming $\mathcal{D}$ is an $(\alpha, \rho, r)$-smooth distribution, for any other threshold $\tau'$,*

$$\frac{\alpha r \cdot (\tau^* - \tau')^2}{2} \leq \mathbb{E}_{\tau \sim \mathcal{D}}[p_q(\tau', \tau) - p_q(\tau^*, \tau)]$$

*Proof.* Assume without loss of generality that $\tau' \leq \tau^*$. Define the probabilities $p_1 = \mathbb{P}(\tau \leq \tau')$, $p_2 = \mathbb{P}(\tau \geq \tau^*)$, and $p_3 = \mathbb{P}(\tau \in [\tau', \tau^*))$, where $\tau \sim \mathcal{D}$. We can compute the expectation $\mathbb{E}[p_q(\tau', \tau) - p_q(\tau^*, \tau)]$ by looking at these three cases separately. When $\tau \leq \tau'$, the difference in loss is $(1 - q)(\tau' - \tau^*)$. Similarly, when $\tau \geq \tau^*$, the difference in pinball loss is $q(\tau^* - \tau')$. Finally, when $\tau' \leq \tau \leq \tau^*$,

$$\begin{aligned}
p_q(\tau', \tau) - p_q(\tau^*, \tau) &= q(\tau - \tau') - (1 - q)(\tau^* - \tau) \\
&= -q(\tau' - \tau^*) - (\tau^* - \tau) \\
&= (1 - q)(\tau' - \tau^*) + (\tau - \tau')
\end{aligned}$$

This difference in loss is dependent on $\tau$, so the conditional expectation in this case is:

$$\mathbb{E}[p_q(\tau', \tau) - p_q(\tau^*, \tau) \mid \tau \in [\tau', \tau^*]] = (1 - q)(\tau' - \tau^*) + \mathbb{E}[\tau - \tau' \mid \tau \in [\tau', \tau^*]]$$

Computing the marginal expectation using the law of total expectation,

$$\begin{aligned}
\mathbb{E}[p_q(\tau', \tau)] - \mathbb{E}[p_q(\tau^*, \tau)] &= p_1(1 - q)(\tau' - \tau^*) + p_2(q)(\tau^* - \tau') + p_3((1 - q)(\tau' - \tau^*) + \mathbb{E}[\tau - \tau' \mid \tau \in [\tau', \tau^*]]) \\
&= p_1(1 - q)(\tau' - \tau^*) + p_2(q)(\tau^* - \tau') + p_3(1 - q)(\tau' - \tau^*) + p_3\mathbb{E}[\tau - \tau' \mid \tau \in [\tau', \tau^*]] \\
&= \mathbb{E}[(\tau - \tau')\mathbf{1}_{\tau' \leq \tau \leq \tau^*}]
\end{aligned}$$

with the final simplication due to $p_1 + p_3 = q$, and $p_2 = 1 - q$, by definition of $\tau^*$. Since $\mathcal{D}$ is $(\alpha, \rho, r)$-smooth, we can obtain a lower-bound on this expectation by taking a discrete sum over $1/r$ pieces of the interval (each of which has probability weight at least $\alpha$). There will be $\lfloor r(\tau^* - \tau') \rfloor$ such intervals, and over the $i$-th such interval, $\tau - \tau' \geq \frac{i-1}{r}$, so we get:

$$\mathbb{E}[p_q(\tau', \tau)] - \mathbb{E}[p_q(\tau^*, \tau)] \geq \frac{\alpha r \cdot (\tau^* - \tau')^2}{2}$$

as desired. The proof for the $\tau^* \leq \tau'$ case is nearly identical. $\square$

**Theorem 3.2.** *Fix a transcript $\Pi_T = \{(\tau_t, \hat{\tau}_t)\}_{t=1}^T$ in a setting without context (i.e. in which there are no observable features $x_t$) and where the sequence of labels is drawn i.i.d. from a fixed distribution, i.e. $\tau_t \sim \mathcal{D}$ for all $t \in [T]$. If $\mathcal{D}$ is $(\alpha, \rho, r)$-smooth, and if $\Pi_T$ has external regret $\gamma$ with respect to the pinball loss $p_q$, then the set of predicted thresholds has marginal coverage error:*

$$|\mathrm{Cov}(\Pi_T) - q| \leq \sqrt{\frac{2\rho(\gamma + 2\epsilon)}{T\alpha}} + \frac{\epsilon}{T}$$

*with probability at least $1 - 6\exp\left(-\frac{\epsilon^2}{2T}\right)$.*

*Proof.* Define the realized loss $L = \sum_{t=1}^T p_q(\hat{\tau}_t, \tau_t)$ and the loss with respect to any fixed threshold $a$, as $L_a = \sum_{t=1}^T p_q(a, \tau_t)$. The regret guarantee tells us that

$$L - L_{\tau^*} \leq \gamma$$

for $\tau^* = \min_{\tau \in [0,1]} \mathbb{E}[L_\tau]$ - this is the $q$-th quantile of the distribution $\mathcal{D}$. For $0 \leq t \leq T$, define the sequence of random variables $X_t = \mathbb{E}[L | \Pi_t]$, adapted to the filtration $\{\Pi_t : t \geq 0\}$. Note that since $\mathbb{E}[X_{t+1} | \Pi_t] = X_t$, this sequence is a martingale. Since $X_0 = \mathbb{E}[L]$ and $X_T = L$, using Azuma's inequality gives us:

$$\mathbb{P}[|L - \mathbb{E}[L]| \geq \epsilon] \leq 2\exp\left(-\frac{\epsilon^2}{2T}\right)$$

Thus we obtain a bound on the difference between expected losses:

$$|\mathbb{E}[L] - \mathbb{E}[L_{\tau^*}]| \leq \gamma + 2\epsilon$$

with probability at least $1 - 4\exp\left(-\frac{\epsilon^2}{2T}\right)$. Using Lemma 3.1 separately for the difference in losses for each time-step,

$$\sum_{t=1}^{T} \frac{\alpha r \cdot (\tau^* - \hat{\tau}_t)^2}{2} \leq \gamma + 2\epsilon \implies \sum_{t=1}^{T} (\tau^* - \tau_t)^2 \leq \frac{2(\gamma + 2\epsilon)}{\alpha r} \tag{2}$$

Now, define for each round $t$ the expected miscoverage $M_t = \mathbb{E}_{\tau \in \mathcal{D}}[1[\hat{\tau}_t \geq \tau]] - q$. Since we know $\tau^*$ achieves the optimal coverage $q$, $M_t = \mathbb{P}(\tau \in [\hat{\tau}_t, \tau^*])$ (or the interval $[\tau^*, \hat{\tau}_t]$), and due to the smoothness condition, this implies that

$$|\hat{\tau}_t - \tau^*| \geq \frac{M_t}{\rho r} \implies (\tau^* - \hat{\tau}_t)^2 \geq \frac{M_t^2}{\rho r}$$

Combining with the inequality from (2), we get:

$$\sum_{t=1}^{T} M_t^2 \leq \frac{2\rho(\gamma + 2\epsilon)}{\alpha} \implies \sum_{t=1}^{T} M_t \leq \sqrt{T}\sqrt{\frac{2\rho(\gamma + 2\epsilon)}{\alpha}}$$

$$\implies \frac{1}{T}\sum_{t=1}^{T} M_t \leq \sqrt{\frac{2\rho(\gamma + 2\epsilon)}{T\alpha}}$$

using Cauchy-Schwarz. Another application of Azuma's inequality tells us that the average expected miscoverage above is more than $\epsilon/T$ away from the realized miscoverage rate with at most probability $2\exp\left(-\frac{\epsilon^2}{2T}\right)$. Taking a union bound over both probabilities, this gives us:

$$|\mathrm{Cov}(\Pi_T) - q| \leq \sqrt{\frac{2\rho(\gamma + 2\epsilon)}{T\alpha}} + \frac{\epsilon}{T}$$

with probability at least $1 - 6\exp\left(-\frac{\epsilon^2}{2T}\right)$. $\qquad\square$

In the stochastic case, we implicitly make the simplifying assumption that the exact $q$-th quantile $\tau^*$ of the distribution $\mathcal{D}$ exists. When we move to the adversarial setting, this is no longer a viable assumption, and we must also consider discretization error. The following lemma is a discretized version of Lemma 3.1.

**Lemma A.1.** *Given a sequence of parameter values $\{\tau_i\}_{i=1}^{T}$, define the sum of pinball losses $L_a = \sum_{i=1}^{T} p_q(a, \tau_i)$ and $L_b = \sum_{i=1}^{T} p_q(b, \tau_i)$ respectively, where $a = \min_{\tau \in \mathcal{A}_n} \sum_{i=1}^{T} p_q(a, \tau_i)$ is the minimizer of the sum of pinball losses over the discretized set $\mathcal{A}_n$, and $b$ is any value in $[0,1]$. If the empirical distribution $\mathcal{D}$ defined by $\{\tau_i\}_{i=1}^{T}$ is $(\alpha, \rho, r)$-smooth, and if $L_b - L_a \leq \gamma$, then $|b - a| \leq \sqrt{\frac{2\gamma}{T\alpha r} + \frac{\rho}{\alpha}\left(\frac{1}{r} + \frac{2}{n}\right)}$.*

*Proof.* Without loss of generality, assume that $a \leq b$. For any fixed $i \in [T]$, consider the difference $\Delta L_i = l_q(b, \tau_i) - l_q(a, \tau_i)$. There are three cases to consider. If $\tau_i < \min\{a, b\}$, then:

$$\Delta L_i = (1 - q)(b - \tau_i - (a - \tau_i)) = (1 - q)(b - a)$$

Similarly, if $\max\{a, b\} \leq \tau_i$, then $\Delta L_i = q(a - b)$. For the third case, consider when $a \leq \tau_i < b$. Then,

$$\Delta L_i = (1 - q)(b - \tau_i) - q(\tau_i - a) = q(a - b) + (b - \tau_i)$$

Let $N_1, N_2$ and $N_3$ be the number of $i \in [t]$ falling into each of these three cases respectively. We first estimate $N_1$; since $a$ minimizes the sum of pinball losses, it must be one of the two grid-points $\mathcal{A}_n$ closest to the value $M$ that minimizes the sum of pinball losses over the continuous interval $[0,1]$ (since the sum of pinball losses is a convex, piece-wise linear function). By definition $M$ must be the value that comes closest to covering $q$ of the probability weight over $\mathcal{D}$. The amount of probability weight on $M$ cannot exceed $\rho$, and $|M - a| \leq 1/n$. By the smoothness condition on $\mathcal{D}$ we have

$|N_1 - qT| \leq \rho T/2 + \rho rT/n$. This implies that $|(N_2 + N_3) - (1-q)T| \leq \rho T/2 + \rho rT/n$. Using $L_b - L_a \leq \gamma$ and rewriting the difference in loss as $\sum_{i=1}^{T} \Delta L_i$,

$$\gamma \geq N_1(1-q)(b-a) + qN_2(a-b) + qN_3(a-b) + \sum_{i:a \leq \tau_i < b} (b - \tau_i)$$

$$= (b-a)(N_1(1-q) - q(N_2 + N_3)) + \sum_{i:a \leq \tau_i < b} (b - \tau_i)$$

$$\geq -(b-a)\left(\frac{\rho T}{2} + \frac{\rho rT}{n}\right) + \sum_{i:a \leq \tau_i < b} (b - \tau_i)$$

$$\geq -\left(\frac{\rho T}{2} + \frac{\rho rT}{n}\right) + \sum_{i:a \leq \tau_i < b} (b - \tau_i)$$

where the second inequality comes from the bounds on $N_1$ and $N_2 + N_3$. Using the smoothness condition on $\mathcal{D}$, we can lower-bound the second term by splitting the interval $[a, b]$ into pieces of length $1/r$, getting

$$\sum_{i:a \leq \tau_i < b} (b - \tau_i) \geq \alpha \sum_{i=1}^{\lfloor r|b-a|\rfloor} \frac{i-1}{r} \geq \frac{T\alpha r(b-a)^2}{2}$$

Rearranging, we get

$$(b-a)^2 \leq \frac{2}{T\alpha r}\left(\gamma + \frac{\rho T}{2} + \frac{\rho rT}{n}\right) \implies |b - a| \leq \sqrt{\frac{2\gamma}{T\alpha r} + \frac{\rho}{\alpha}\left(\frac{1}{r} + \frac{2}{n}\right)}$$

$\square$

**Theorem 3.5.** *Fix a transcript $\Pi_T = \{(x_t, \tau_t, \hat{\tau}_t)\}_{t=1}^{T}$. If $\Pi_T$ has swap regret $\gamma$ with respect to the pinball loss $p_q$, and the empirical distribution $\mathcal{D}_\tau$ defined by the set $\{\tau_t\}_{t:\hat{\tau}=\tau}$ is $(\alpha, \rho, r)$-smooth for each $\tau \in \mathcal{A}_n$, then the set of predicted thresholds satisfies threshold-calibrated coverage at the level $q$:*

$$|Cov(\Pi_T, G_\tau) - q| \leq \frac{\rho}{2} + \frac{\rho r}{n} + \sqrt{\frac{2\gamma}{T_\tau \alpha r} + \frac{\rho}{\alpha}\left(\frac{1}{r} + \frac{2}{n}\right)}$$

*Proof.* Since each predicted value $\hat{\tau}_t$ is in $\mathcal{A}_n$, we can rewrite regret via the separate contributions over each prediction value:

$$r(\Pi_t, p_q, \phi) = \sum_{\tau \in \mathcal{A}_n} \underbrace{\sum_{t:\hat{\tau}_t = \tau} p_q(\hat{\tau}_t, \tau_t) - p_q(\phi(\hat{\tau}_t), \tau_t)}_{r_{\tau,\phi}}$$

Define the swap function $\phi_m$ that, for each $\tau \in \mathcal{A}_n$, is defined as:

$$\phi_m(\tau) = \min_{\tau' \in \mathcal{A}_n} \sum_{t:\hat{\tau}=\tau} p_q(\tau', \tau_t)$$

as well as the loss minimizer mapping $M : \mathcal{A}_n \to [0,1]$:

$$M(\tau) = \min_{\tau' \in [0,1]} \sum_{t:\hat{\tau}=\tau} p_q(\tau', \tau_t)$$

Note that by definition, since $M(\tau)$ minimizes the sum of pinball losses, it is the $q$-th quantile of the empirical distribution $\mathcal{D}_\tau$ over the set $\{\tau_t\}_{t:\hat{\tau}=\tau}$. Further, since the sum of pinball losses (as a function of the first argument) is a convex, piece-wise linear function, $\phi_m(\tau)$ must be one of the two closest grid-points in $\mathcal{A}_n$ to $M(\tau)$, i.e. we have $|M(\tau) - \phi_m(\tau)| \leq 1/n$. Since $r_{\tau,\phi_m} \geq 0$ for each $\tau \in \mathcal{A}_n$, a total swap-regret of $\gamma$ implies that $r_{\tau,\phi_m} \leq \gamma$ for each $\tau$. Using Lemma A.1,

$$|\phi_m(\tau) - \tau| \leq \sqrt{\frac{2\gamma}{T_\tau \alpha r} + \frac{\rho}{\alpha}\left(\frac{1}{r} + \frac{2}{n}\right)}$$

where we define $T_\tau = \sum_{t \in [T]} \mathbf{1}[\hat{\tau}_t = \tau]$. Due to the $(\alpha, \rho, r)$-smoothness condition over $\mathcal{D}_\tau$, the amount of probability weight on $M(\tau)$ cannot exceed $\rho$, and so the number of values $N_\tau$ in $\{\tau_t\}_{t:\hat{\tau}=\tau}$ that $M(\tau)$ equals or exceeds satisfies $qT_\tau - \rho T_\tau / 2 \le N_\tau \le qT_\tau + \rho T_\tau / 2$. Finally, using the bound on $|M(\tau) - \tau|$ along with the smoothness condition, the number of values in the set $\{t : \hat{\tau} = \tau\}$ between $M(\tau)$ and $\tau$ cannot exceed $T_\tau \cdot \rho r \cdot \left( \frac{1}{n} + \sqrt{\frac{2\gamma}{T_\tau \alpha r} + \frac{\rho}{\alpha}\left(\frac{1}{r} + \frac{2}{n}\right)} \right)$. Using the upper bounds of the inequalities,

$$\sum_{t:\hat{\tau}=\tau} \mathbf{1}[\tau_t \le \tau] \le qT_\tau + \frac{\rho T_\tau}{2} + \frac{\rho r T_\tau}{n} + \sqrt{\frac{2\gamma}{T_\tau \alpha r} + \frac{\rho}{\alpha}\left(\frac{1}{r} + \frac{2}{n}\right)}$$

Notice that the left hand side equals $\mathrm{Cov}(\Pi_T, G_\tau)$, where $G_\tau$ is the binary group including all time-steps $t$ for which $\hat{\tau}_t = \tau$. Thus, performing the same steps using the lower bounds of the inequality and dividing by $T_\tau$,

$$|\mathrm{Cov}(\Pi_T, G_\tau) - q| \le \frac{\rho}{2} + \frac{\rho r}{n} + \sqrt{\frac{2\gamma}{T_\tau \alpha r} + \frac{\rho}{\alpha}\left(\frac{1}{r} + \frac{2}{n}\right)}$$

$\square$

**Theorem 3.6.** *Fix a transcript $\Pi_T = \{(x_t, \tau_t, \hat{\tau}_t)\}_{t=1}^T$. If $\Pi_T$ has threshold-calibrated coverage with coverage error $\gamma$ (at desired coverage rate $q$), and $\mathcal{D}_\tau$ defined by $\{\tau_t\}_{t:\hat{\tau}=\tau}$ is $(\alpha, \rho, r)$-smooth for each $\tau \in \mathcal{A}_n$, then the transcript also has swap regret with respect to the loss $p_q$, such that:*

$$r(\Pi_t, p_q, \phi) \le \frac{T\gamma^2 \rho}{\alpha^2 r}$$

*for each $\phi \in \Phi$, the collection of all strategy modification rules for action set $\mathcal{A}_n$.*

*Proof.* Fix a threshold $\tau \in \mathcal{A}_n$. Let $M(\tau) = \min_{a \in [0,1]} |\mathrm{Cov}(\Pi_T, G_\tau) - q|$ where $G_\tau$ is the binary group including all time-steps for which the predicted threshold was $\tau$. Note that since by definition $M(\tau)$ is the $q$-th quantile of the empirical distribution $\mathcal{D}_\tau$, it is also the value $a$ that minimizes the sum of pinball losses $\sum_{t=1}^T \mathbf{1}[\hat{\tau}_t = \tau] \cdot p_q(a, \tau_t)$. We assume without loss of generality that this exact $q$-th quantile over $\mathcal{D}_\tau$ exists - since any other value would get worse coverage (at the rate $q$), $\tau$ achieving comparable performance to the true minimizer implies it achieves the same (or better) performance even if the exact $q$-th quantile does not exist. By the coverage error guarantee,

$$\mathbb{P}_{\tau \in \mathcal{D}_\tau}(\tau \in [M(\tau), \tau]) \le \gamma$$

or instead $[\tau, M(\tau)]$, based on their ordering. Using the smoothness condition, we have:

$$|\tau - M(\tau)| \le \frac{\gamma}{\alpha r}$$

Thus the regret with respect to the best action in hindsight $M(\tau)$ over the subsequence where only prediction $\tau$ is made can be bound:

$$r_\tau = \max_{a \in \mathcal{A}_n} \sum_{t:\hat{\tau}_t = \tau} p_q(\tau, \tau_t) - p_q(a, \tau_t)$$

$$\le \sum_{t:\hat{\tau}_t = \tau} p_q(\tau, \tau_t) - p_q(M(\tau), \tau_t) \le \left(\frac{\gamma}{\alpha r}\right)^2 \rho r$$

Assume without loss of generality that $M(\tau) \le \tau$. Define the variables $N_1, N_2$ and $N_3$ as the number of thresholds in the set $\{\tau_t\}_{t:\hat{\tau}_t = \tau}$ less than or equal to $M(\tau)$, in the interval $(M(\tau), \tau]$, and greater than $\tau$ respectively. Since $M(\tau)$ is exactly the $q$-th quantile, $N_1 = qT_\tau$ and $N_2 + N_3 = (1-q)T_\tau$, where we define $T_\tau = \sum_{t=1}^T \mathbf{1}[\hat{\tau} = \tau]$. We can rewrite the difference in pinball loss by dividing into these three categories, as in the proof for Lemma $A.1$, to get:

$$\sum_{t:\hat{\tau}_t = \tau} p_q(\tau, \tau_t) - p_q(M(\tau), \tau_t) = N_1(1-q)(\tau - M(\tau)) - qN_2(\tau - M(\tau)) + qN_3(\tau - M(\tau)) + \sum_{i:a \le \tau_i < b, \hat{\tau}_i = \tau} (\tau - \tau_i)$$

$$= (\tau - M(\tau))(N_1(1-q) - q(N_2 + N_3)) + \sum_{i:a \le \tau_i < b, \hat{\tau}_i = \tau} (\tau - \tau_i)$$

$$\le N_3(\tau - M(\tau)) \le T_\tau \frac{\gamma \rho r}{\alpha r} \cdot \frac{\gamma}{\alpha r} = T_\tau \frac{\gamma^2 \rho}{\alpha^2 r}$$

using the smoothness condition and the bound on $|M(\tau) - \tau|$ to bound the value of $N_3$. Thus the maximal regret with respect to the best action (on each subsequence defined by a fixed threshold prediction) is bounded. Summing across all $\tau \in \mathcal{A}_n$,

$$r(\Pi_t, p_q, \phi) \leq \frac{T\gamma^2 \rho}{\alpha^2 r}$$

$\square$

**Theorem 3.7.** *Fix a transcript $\Pi_T$, and a set of binary groups $\mathcal{G}$. If $\Pi_T$ has groupwise swap regret $\gamma$ with respect to the pinball loss $p_q$, and the empirical distributions $\mathcal{D}_{G,\tau}$ defined by the set $\{\tau_t\}_{t:\hat{\tau}=\tau, t \in G}$ are $(\alpha, \rho, r)$-smooth, then the set of predicted thresholds satisfies multivalid coverage at the level $q$ such that*

$$|\mathrm{Cov}(\Pi_T, H_{G,\tau}) - q| \leq \frac{\rho}{2} + \frac{\rho r}{n} + \sqrt{\frac{2\gamma}{T_{G,\tau}\alpha r} + \frac{\rho}{\alpha}\left(\frac{1}{r} + \frac{2}{n}\right)}$$

*for each group $H_{G,\tau}$, defined as $H_{G,\tau}(\Pi_t, x_t, \hat{\tau}_t) = G(\Pi_t, x_t, \hat{\tau}_t) \cdot \mathbf{1}[\hat{\tau}_t = \tau]$.*

*Proof.* We follow a nearly identical approach to Theorem 3.5. Fix a group $G \in \mathcal{G}$. The groupwise regret with respect to $G$ can be rewritten via the separate contributions over each prediction value:

$$r(\Pi_t, p_q, \phi, G) = \sum_{\tau \in \mathcal{A}_n} \underbrace{\sum_{t:\hat{\tau}_t = \tau} (p_q(\hat{\tau}_t, \tau_t) - p_q(\phi(\hat{\tau}_t), \tau_t)) \cdot G(\Pi_t, x_t, \hat{\tau}_t)}_{r_{G,\tau,\phi}}$$

Defining $\phi_m$ and $M$ as in Theorem 3.5, we get that

$$|M(\tau) - \tau| \leq \frac{1}{n} + \sqrt{\frac{2\gamma}{T_{G,\tau}\alpha r} + \frac{\rho}{\alpha}\left(\frac{1}{r} + \frac{2}{n}\right)}$$

where we define $T_{G,\tau} = \sum_{t \in [T]} \mathbf{1}[\hat{\tau}_t = \tau] \cdot G(\Pi_t, x_t, \hat{\tau}_t)$. Since the amount of probability weight on $M(\tau)$ cannot exceed $\rho$, we get an upper-bound on the group conditional coverage:

$$\sum_{t:\hat{\tau}=\tau} \mathbf{1}[\tau_t \leq \tau] \cdot G(\Pi_t, x_t, \hat{\tau}_t) \leq qT_{G,\tau} + \frac{\rho T_{G,\tau}}{2} + \frac{\rho r T_{G,\tau}}{n} + \sqrt{\frac{2\gamma}{T_{G,\tau}\alpha r} + \frac{\rho}{\alpha}\left(\frac{1}{r} + \frac{2}{n}\right)}$$

and then a similar lower-bound. So,

$$|\mathrm{Cov}(\Pi_T, H_{G,\tau}) - q| \leq \frac{\rho}{2} + \frac{\rho r}{n} + \sqrt{\frac{2\gamma}{T_{G,\tau}\alpha r} + \frac{\rho}{\alpha}\left(\frac{1}{r} + \frac{2}{n}\right)}$$

where $H_{G,\tau}$ is the group defined such that $H_{G,\tau}(\Pi_t, x_t, \hat{\tau}_t) = G(\Pi_t, x_t, \hat{\tau}_t) \cdot \mathbf{1}[\hat{\tau}_t = \tau]$. $\square$

**Theorem 3.8.** *Fix a transcript $\Pi_T$, and a set of binary groups $\mathcal{G}$. If $\Pi_T$ has multivalid coverage with coverage error $\gamma$ (at desired coverage rate $q$), and $\mathcal{D}_{G,\tau}$ is $(\alpha, \rho, r)$-smooth for each $\tau \in \mathcal{A}_n, G \in \mathcal{G}$, then the transcript also has groupwise swap regret with respect to the loss $p_q$, such that $r(\Pi_t, p_q, \phi, G) \leq \frac{T_G\gamma^2 \rho}{\alpha^2 r}$ for each $\phi \in \Phi$, the collection of all strategy modification rules for action set $\mathcal{A}_n$, where $T_G$ is the size of group $G$.*

*Proof.* Fix a threshold $\tau \in \mathcal{A}_n$ and a group $G \in \mathcal{G}$. Using exactly the approach in the proof for Theorem 3.6, we can bound the sum of difference in loss, over the subsequence for which group $G$ is active and the predicted threshold was $\tau$:

$$\sum_{t:\hat{\tau}_t = \tau} (p_q(\tau, \tau_t) - p_q(M(\tau), \tau_t)) \cdot G(\Pi_t, x_t, \hat{\tau}_t) \leq T_{G,\tau} \frac{\gamma^2 \rho}{\alpha^2 r}$$

where $T_{G,\tau} = \sum_{t \in [T]} \mathbf{1}[\hat{\tau}_t = \tau] \cdot G(\Pi_t, x_t, \hat{\tau}_t)$. Summing over all $\tau \in \mathcal{A}_n$, we get

$$r(\Pi_t, p_q, \phi, G) \leq \frac{T_G\gamma^2 \rho}{\alpha^2 r}$$

$\square$

**Theorem 4.1.** *For the parametrization of FTRL given in Algorithm 1 with regularization function $R : \mathbb{R}^d \to \mathbb{R}$, for any target coverage rate $q$ and any $T$ the resulting transcript $\Pi_T$ is guaranteed to satisfy groupwise coverage for groups $G_i$ ($i \in [k]$) at the rate:*

$$|\text{Cov}(\Pi_T, G_i) - q| \leq \frac{||\nabla R(\theta_{T+1})||_\infty}{T_i}$$

*Proof.* Pinball loss is convex, and so to apply FTRL, we feed the algorithm the linear surrogate loss $\ell(\theta, \tau_t) \doteq \langle \theta, \nabla_\theta p_q(\langle \theta_t, g_t \rangle, \tau_t) \rangle$. We can compute:

$$\ell(\theta, \tau_t) = \begin{cases} -q \langle \theta, g_t \rangle, & \text{if } \tau_t > \langle \theta_t, g_t \rangle, \\ (1-q) \langle \theta, g_t \rangle, & \text{if } \tau_t \leq \langle \theta_t, g_t \rangle. \end{cases}$$

The gradient of the loss at round $t$ with respect to $\theta$ is therefore:

$$\nabla_\theta \ell(\theta, \tau_t) = \begin{cases} -q \, g_t, & \text{if } \tau_t > \langle \theta_t, g_t \rangle, \\ (1-q) \, g_t, & \text{if } \tau_t \leq \langle \theta_t, g_t \rangle. \end{cases}$$

FTRL with regularizer $R$ plays the action $\theta_t$ at round $t$ that solves:

$$\theta_t = \arg\min_\theta \sum_{s=1}^{t-1} \ell(\theta, \tau_s) + R(\theta)$$

First order optimality conditions imply that:

$$\sum_{s=1}^{t-1} \nabla_\theta \ell(\theta_t, \tau_s) + \nabla R(\theta_t) = 0$$

Or equivalently,

$$\begin{aligned} \nabla R(\theta_t) &= \sum_{s:\tau_s > \langle \theta_s, g_s \rangle} q g_s + \sum_{s:\tau_s \leq \langle \theta_s, g_s \rangle} (q-1) g_s \\ &= \sum_{s=1}^{t-1} g_s(q - 1[\tau_s \leq \hat{\tau}_s]) \end{aligned}$$

Hence we can bound the miscoverage rate for every group $i$ at time $T$ can be bounded as:

$$|\text{Cov}(\Pi_T, G_i) - q| \leq \frac{||\nabla R(\theta_{T+1})||_\infty}{T_i}$$

$\square$

**Lemma 5.1.** *Running Algorithm 2 for any number of rounds $T$ with a coverage target of $q$ for any set of $k$ group functions, we achieve groupwise miscoverage bounded by the following function of $\theta_{T+1}$:*

$$|\text{Cov}(\Pi_T, G_i) - q| \leq \frac{||\theta_{T+1}||_\infty}{T_i \eta}$$

*Proof.* Algorithm 2 is an instantiation of follow the regularized leader as analyzed in Theorem 4.1 with regularization function $R(\theta) = \frac{1}{2\eta} ||\theta||^2$. We can compute $\nabla R(\theta_{T+1}) = \frac{1}{\eta} \cdot \theta_{T+1}$. Plugging this into Theorem 4.1 gives the stated bound. $\square$

**Lemma 5.2.** *When Algorithm 2 is run with step-size $\eta \in (0, 1]$, for any collection of $k$ group functions, any coverage target $q \in (0, 1)$, and every $T$, the iterate $\theta_{T+1}$ has norm bounded as:*

$$||\theta_{T+1}||_\infty \leq \mathcal{O}(\sqrt{\eta T(\eta k + 1)})$$

*Proof.* First, note that since the non-conformity scores are assumed to be bounded in $[0, 1]$, we must have for every $t$ that $\tau_t \in [0, 1]$. So, if $\langle \theta_t, g_t \rangle < 0$, we must also have $\langle \theta_t, g_t \rangle \leq \tau_t$ which triggers Update A. Similarly, whenever $\langle \theta_t, g_t \rangle \geq 1$, this necessarily triggers update B. Said another way: if the update A was triggered at round $t$ we know that $\langle \theta_t, g_t \rangle < 1$, whereas if update B was triggered, we know that $\langle \theta_t, g_t \rangle \geq 0$. We consider these two cases separately.

**Case 1 (Update A triggered):** We can compute

$$\|\theta_{t+1}\|_2^2 = \|\theta_t\|_2^2 + \eta^2 q^2 \|g_t\|_2^2 + 2\eta q \langle \theta_t, g_t \rangle$$
$$\leq \|\theta_t\|_2^2 + \eta^2 q^2 k + 2\eta q$$

**Case 2 (Update B triggered):** Similarly,

$$\|\theta_{t+1}\|_2^2 = \|\theta_t\|_2^2 + \eta^2 (1-q)^2 \|g_t\|_2^2 - 2\eta(1-q)\langle \theta_t, g_t \rangle$$
$$\leq \|\theta_t\|_2^2 + \eta^2 (1-q)^2 k$$

As initially $\|\theta_1\|_2 = 0$, we obtain:

$$\|\theta_{T+1}\|_2^2 \leq T \left( \eta^2 k \max\{q, 1-q\}^2 + 2\eta q \right)$$
$$\leq T\eta \left( \eta k \max\{q, 1-q\}^2 + 2q \right)$$

This immediately gives us a bound on the $L_\infty$ norm:

$$\|\theta_{T+1}\|_\infty \leq \sqrt{T\eta} \sqrt{\eta k \max\{q, 1-q\}^2 + 2q}$$

$\square$

**Theorem 5.4.** *Let $k = \eta = 1$ and pick any coverage target $q \in (0, 1)$. The sequence of 1-dimensional weighting functions $g_t = \frac{1}{2\sqrt{t-1}}$ together with thresholds $\tau_t = 1$ causes Algorithm 2 to produce parameter vector $\theta_{T+1} \in \Omega(\sqrt{T})$.*

*Proof.* Assume that $g_1 = 0$. Since $\tau_t = 1$, whenever $\theta_t \cdot g_t < 1$ we will trigger update A. Assume all rounds up through round $t-1$ triggered update $A$, in which case $\theta_t = \eta \cdot q \cdot \sum_{k=1}^{t-1} g_k < 2\sqrt{t-1}$. But because we set $g_t = \frac{1}{2\sqrt{t-1}}$ we have that $\theta_t \cdot g_t < 1$, once again triggering update $A$. Inductively, update $A$ is thus triggered at every round, and so we have that $\theta_{T+1} = \eta \cdot q \cdot \sum_{k=1}^{T} g_k = \Omega(\sqrt{T})$. $\square$

# B. Additional Related Work

In a parallel line of work, Gupta et al. (2022) introduced the problem of online uncertainty quantification in the form of mean, variance, and quantile estimation, using techniques deriving from the online calibration literature (Foster & Vohra, 1998). Bastani et al. (2022) gave a refinement of their quantile calibration technique to give an online conformal prediction method that gave conditional guarantees of various sorts. Coverage bounds from algorithms of this sort follow from quantile-calibration arguments.

(Foygel Barber et al., 2021) consider the problem of group conditional coverage in conformal prediction and propose running separate algorithms for each group, and for examples that are in multiple groups, using the most conservative threshold amongst each of the groupwise algorithms. (Jung et al., 2021) give the first non-conservative method for getting groupwise coverage for intersecting groups, by adapting ideas from multicalibration (Hébert-Johnson et al., 2018) to calibrate to *moments* of the score function, conditional on group membership. (Gupta et al., 2022) give algorithms for group-conditional quantile multicalibration, and show how this can be used to give tight "multivalid" confidence intervals. (Bastani et al., 2022) and (Jung et al., 2023) apply these ideas explicitly to conformal prediction. (Gibbs et al., 2025) give a variant of the algorithm from (Jung et al., 2023) which gives coverage guarantees in expectation over the calibration set, rather than PAC-style guarantees as in (Jung et al., 2023).

The characterization we give of threshold calibrated coverage by swap regret bounds on the pinball loss mirrors an equivalence between swap regret on the squared loss and (mean) calibration (Foster & Vohra, 1998; 1999). More generally the connection between calibration of different distributional quantities and their corresponding "elicitation functions" was made by (Noarov & Roth, 2023).

# C. Experiments

Here we provide additional details on the set-up for each experiment. We closely follow the steps described in (Bastani et al., 2022), which performed experiments on the same datasets to measure the performance of their algorithm MVP for group-conditional coverage guarantees.

## C.1. Time Series Data

We replicate the prediction task described first in (Gibbs & Candes, 2021), for testing the ACI algorithm's ability to achieve marginal coverage, which uses AMD stock market data from the WSJ daily price across years 2000-2020. The dataset gives price points $\{p_t\}_{t=1}^{T}$ of the stock for $T = 5283$. Using this data, we compute the daily return $r_t$, defined as $r_t = \frac{p_t - p_{t-1}}{p_{t-1}}$, which correspondingly defines the daily realized volatility $v_t = r_t^2$. The task is to predict this volatility. Using the predictive model GARCH ((Bollerslev, 1986)), which makes a prediction of the volatility $\hat{v}_t$, the scoring function used on day $t$ is $f_t(x, y) = \frac{|y - \hat{v}_t|}{\hat{v}_t}$, normalized to ensure scores are always in the range $[0, 1]^2$. Then, as in (Bastani et al., 2022), we define the collection of 20 groups $\{G_i\}_{i=1}^{20}$, where $G_i$ includes all time-steps $t$ for which $t \equiv 0 \pmod{i}$, and introduce artificial noise to group data in the following way - for each time-step $t$, we add noise $\mathcal{N}(0, \hat{\sigma}_r)$ to the return value $t$ for each group in $G_i$ that $t$ is included in, where $\hat{\sigma}_r$ is the standard deviation of the original return sequence. We run both GCACI and MVP on this data, asking for a desired group conditional coverage level of $q = 0.9$. In (Bastani et al., 2022), they show that MVP achieves the desired group conditional coverage while ACI is unable to. Here, we see that GCACI not only achieves group conditional coverage, but converges at much quicker rates than MVP.

## C.2. Synthetic Distribution Shift (UCI Airfoil Data)

We run both MVP and GCACI on the airfoil dataset from the UCI Machine Learning Repository (Dua & Graff, 2017), which consists of 1503 instances of NASA airfoil blades; the task is to predict the Scaled Sound Pressure Level (SSPL). In (Bastani et al., 2022), they compare against the performance of the weighted split conformal prediction algorithm of (Tibshirani et al., 2019), and test only for marginal coverage. Following their approach, we use 25% of the data to train a linear regression model $g : \mathcal{X} \to \mathbb{R}$, which defines the scoring function $f(x, y) = |g(x) - y|$. Another 25% of the data is used as is, and the final 50% of the data is sampled (with replacement) using exponential tilting - each datapoint $x$ is drawn with probability proportional to $\exp(\langle x, \beta \rangle)$, where we set $\beta = (-1, 0, 0, 0, 1)$ as in (Tibshirani et al., 2019) and (Bastani et al., 2022), representing synthetic covariate shift. The test set is sequenced such that the original (unshifted) data comes first, followed by the shifted data. Then, as in the previous section, we define a set of six groups $\{G_i\}_{i=1}^{6}$ where membership is again defined by time-step, i.e. $G_i$ includes all time-steps for which $t \equiv 0 \pmod{i}$. Both algorithms are run with a desired coverage rate $q = 0.9$.

## C.3. Natural Distribution Shift (Folktables)

Finally, we compare performance of MVP and GCACI on a distribution shift problem using 2018 Census data from the Folktables repository (Ding et al., 2021). The task involves predicting individuals' income. We use census data from two different states (California & Pennsylvania) and sample 0.2 of both states to get a test set with $N = 52794$ data points. The data is sequenced with all CA datapoints first, giving us unknown distribution shift from a natural source. A quantile regression model $h : \mathcal{X} \to \mathbb{R}$ is trained on 50% of the remaining California data, defining the fixed scoring function $f(x, y) = |h(x) - y|$. We define 12 total groups, over all nine codes for race available in the Folktables dataset, two groups for sex, as well as the group including all data points. In (Bastani et al., 2022), four of the race groups are omitted due to being very small fractions of the overall dataset - we include even these small-sized groups to illustrate that GCACI is able to converge quickly even for such groups. We run both algorithms with a desired coverage rate of $q = 0.9$.

---

[2]Note that though the feature vector $x$ is included generically here as an argument in the non-conformity score, the GARCH model typically uses only past volatility data to make predictions for the next time-step in the sequence.

