# OpenReview forum: "The Relationship Between No-Regret Learning and Online Conformal Prediction"
_ICML.cc/2025/Conference — ICML 2025 poster_

### Official Review · Reviewer_b1E4 · 2025-03-08

**Overall Recommendation:** 3

**Summary:**

The paper investigates the relationship between coverage and various definitions of regret in online learning. The results of this investigation are as follows:
- Sublinear regret with respect to the pinball loss implies coverage guarantees if the data sequence is i.i.d. and a smoothness condition holds on the distribution of the data.
- Sublinear swap regret  implies (approximate) threshold-calibrated coverage when the empirical distribution of the threshold is smooth.
- Sublinear  Groupwise swap regret can be linked to (approximate) group-wise coverage if the empirical distribution of the threshold is smooth.

Algorithmically, the author proposes group conditional ACI, a variant of ACI, which guarantees coverage for different groups in the covariate space.

**Claims And Evidence:**

Yes, all claims are supported by proof and theorems.

**Essential References Not Discussed:**

N/A

**Experimental Designs Or Analyses:**

I have read the description of the experimental set-up and it appears to be reasonable. However,  doing the experiments with datasets with more natural membership conditions would be more instructive.

**Methods And Evaluation Criteria:**

Yes, the authors consider simple benchmarks where group membership is accounted for by differentiating samples over time or features. Except for the last example, the group membership definitions are artificially defined. Datasets with more natural membership conditions would be more instructive.

**Other Comments Or Suggestions:**

As mentioned in the previous section, I would suggest better contextualizing the theorems, explaining their practical relevance, and highlighting more clearly how the theoretical results are used to analyze G-ACI in a way that could not have been done before.

**Other Strengths And Weaknesses:**

The paper’s theoretical contributions are sound; however, it is not clear how one benefits from the derived guarantees. More specifically, I don’t see how they refine the analysis of existing online conformal calibration algorithms or lead to new ones. I would suggest elaborating on the implications of each result in the context of online calibration. As they are presented now, I have the impression that the results lack sufficient context and fail to explain their practical relevance to the field of online calibration.

The criticism above applies to Section 5, in which G-ACI is introduced. In fact, it appears that the proposed algorithm is derived using standard techniques that are independent of the results developed in the preceding sections.

**Questions For Authors:**

- Theorems 3.5 to 3.8 make some assumptions about the smoothness of the empirical distribution of $\tau$ values at specific intervals. Does this imply an assumption on the distribution of the sequence $(x_t, y_t)$? If so, can this be elucidated? does this means that this coverage guarantees are not adversarial as normally assumed in online CP?
- In the abstract, the perturbed leader algorithm is mentioned but the connection between this algorithm and online CP is not clarified in the main text.


Update after rebuttal: I have decided to raise my original score to weak accept.

**Relation To Broader Scientific Literature:**

I think the paper does a good job explaining what are the existing work in the field.

**Theoretical Claims:**

I have checked the proof on the coverage guarantees of G-ACI and those about the relationship between swap regret and coverage.

---

> ### Author Rebuttal · Authors · 2025-03-31
>
> Thank you for your thoughtful feedback! We address your main concerns here:
>
> 1. **Benefits / Relevance of our work:** In regards to the guarantees of our paper, we wish to clarify a misconception the reviewer may have. Our goal in this paper is to elucidate the relationship between regret and coverage in two separate ways:
>
>     * What is the direct relationship between the *properties* of “no-regret” and “marginal/group-conditional coverage” in conformal prediction. This is algorithm-agnostic --- we are interested in when one property of an algorithm generically implies another.
>
>     * What are algorithms that are able to simultaneously achieve both kinds of properties, even when the properties are not (approximately) equivalent. This is algorithm-specific, since when the properties are not equivalent, our analysis needs to be specific to the algorithm itself.
>
>     Our goal in this paper is to make explicit which connections between regret and coverage are connected through the first point and which are connected through the second. To our knowledge, apart from concurrent work in [1], no other work in either the no-regret or the online conformal prediction literature has focused on distinguishing between these points as well as substantiating when the properties do not imply one another.
>
>     Our discussion in Sections 4 & 5 (including the introduction of GCACI) examines the second point. Though external regret wrt pinball loss does *not* imply coverage (as we show in Section 3), a general class of FTRL algorithms (which have been known for several decades to achieve external regret) are *also* able to achieve coverage guarantees bounded by the norm of the regularizer’s value. This is an insight not found in prior work, and it is through this connection that we instantiate the GCACI algorithm. The prior work related to this algorithm (ACI and its later variants) all apply only to the marginal, not the group conditional case. We will add discussion in our results section to explain this with more clarity.
>
>     As far as practical benefits, we show in our experiments that GCACI, in addition to being more lightweight, achieves much faster convergence rates than the MVP algorithm from  [2] which satisfies the stronger threshold-calibration guarantee. Previously, the only two families of online conformal prediction algorithms (MVP, and ACI and its variants) differed in multiple respects: the ACI algorithms obtained faster convergence and were more lightweight, but offered weaker guarantees than MVP in two respects: they were not threshold calibrated, and they did not offer group conditional guarantees. Here we show that we can obtain lightweight algorithms with fast convergence by giving up only on threshold calibration, while keeping group conditional validity --- this was not previously known.
>
> ####
>
> 2. **Smoothness requirement:** Smoothness is a mild condition that we can enforce if necessary by perturbing the scores chosen by the adversary (which implicitly perturbs the threshold) by a value drawn from a uniform distribution $[-\epsilon, \epsilon]$. This doesn’t require us to make assumptions on the adversary.
>
> 	We also note that Theorems 3.5 through 3.8 are making direct connections between no regret and coverage, and it is not possible to make this connection without some kind of smoothness guarantee - we address this in more detail in point (1) with reviewer **8yW2**.
>
> ####
>
> 3. **Follow-the-perturbed-leader mention in abstract:** This is a typo; it should be follow-the-regularized leader instead. Oops! Thanks for pointing it out!
>
> ####
>
> 4. **More natural definition of group-membership:** Our experiments are of course synthetic, and you are completely right that in some of them the groups are not “natural”. Our goal in these experiments is simply to compare the convergence rates of coverage, and for this purpose we think that the results are agnostic to the meaning of the group functions. In general, we agree that it is more interesting to define groups based on features we wish to eliminate disparity between, as we do in the Folktables experiments.
>
> 	The time-series data does not come with features, so we found the most natural way to define groups was as a function of the time-step. For the UCI Airfoil data, all features are numerical variables measuring parts of the airfoils - we did not find there was any meaningful way to define groups through these variables.
>
> ####
>
> [1] Angelopoulos, Anastasios N, et al. “Gradient Equilibrium in Online Learning: Theory and Applications.” ArXiv.org, 2025, arxiv.org/abs/2501.08330
>
> [2] Bastani, Osbert, et al. “Practical Adversarial Multivalid Conformal Prediction.” ArXiv.org, 2022, arxiv.org/abs/2206.01067.

---

> > ### Comment · Reviewer_b1E4 · 2025-04-07
> >
> > Thank you for the detailed response and for clarifying the scope of your results and the smoothness condition. I appreciate the insights provided between the relationship between regret and coverage; however, I see that reviewer obBc shares my concern/misconception regarding the interpretation of the presented results and what practical conclusions one might draw from them.
> >
> > I agree that the results are algorithm agnostic; however would it be possible  to present, for each result, a known algorithm that satisfies the stated properties in online convex optimization, along with the corresponding implications for calibration? For example, elaborating on how this existing algorithm would look like for calibration or what guarantees it would have (similarly as done for GCACI)

---

> > > ### Author Response · Authors · 2025-04-07
> > >
> > > Hi! Thanks for continuing to engage. Sure, we're happy to flesh out the algorithmic implications of our swap regret results.
> > >
> > > The best rates for swap regret are obtained by the algorithm of Blum and Mansour, 2007. The algorithm is parameterized over a set of $n$ actions, which for us will be discrete points on the unit interval $A = (1/n, 2/n, \ldots, \frac{n-1}{n}, 1)$. The algorithm runs n copies of multiplicative weights, each of which maintains a distribution over the $n$ actions in $A$. The algorithm then "stacks" these $n$ distributions into an $n\times n$ matrix $M$, and computes the top eigenvector $p \in \mathbb{R}^n$ of this matrix --- i.e. a vector $p$ such that $Mp = p$. Since it is a stochastic matrix, $p$ has eigenvalue $1$ and $p$ can itself be normalized to be a probability distribution over $A$. The algorithm selects an action according to the distribution $p$. When the conformity score is observed, the algorithm can compute a loss vector $\ell \in \mathbb{R}^n$ (corresponding to the pinball loss of each discrete threshold in $A$). Losses are then fed back to each copy of multiplicative weights; the loss vector fed to copy $i$ is scaled by $p_i$: $\ell^i = p_i \cdot \ell$.
> > >
> > > Generically the Blum/Mansour algorithm gets a swap regret bound of $R(T) \leq O\left(\sqrt{Tn}\right)$. In our case, as pinball loss is Lipschitz, the best threshold in hindsight in the discrete set $A$ might have cumulative pinball loss that is higher than the best threshold in hindsight by as much as $T/n$, and so in terms of our discretization threshold, the swap regret bound with respect to pinball loss will be $O\left(\frac{T}{n} + \sqrt{Tn}\right)$. Choosing the value of $n$ to optimize this bound ($n = T^{1/3}$) we get a swap regret bound to pinball loss of $R(T) \leq O(T^{2/3})$.
> > >
> > > We can now plug this swap regret bound into Theorem 3.5. Doing so, we find that (under the assumption that the distribution is $(\alpha, \rho, r)$-smooth), for any threshold $\tau \in A$ that has been used $T_\tau$ many times, conditional on having used $T_\tau$, the coverage error (i.e. the deviation from the realized coverage rate and the target coverage rate $q$) is bounded by:
> > > $$O\left(\frac{\rho}{2} + \frac{\rho r}{n} + \sqrt{\frac{T^{2/3}}{T_\tau \alpha r}} \right)$$
> > >
> > > Thus, for any threshold $\tau$ played a constant fraction of rounds (in fact, played at least an $\omega(1/T^{1/3})$ fraction of rounds suffices), the coverage converges to $O\left(\frac{\rho}{2} + \frac{\rho r}{n}\right)$, a baseline error rate depending only on the smoothness parameters of the distribution. Its not hard to see that predictions on a finite grid at discretization $1/n$ cannot drive the threshold coverage error below this --- the prior MVP algorithm has a similar term dependent on the smoothness parameters.
> > >
> > >
> > > We're happy to add this discussion/calculation to the paper. As we note, we view the connection between swap regret and threshold calibrated coverage mainly as an important characterization to understand the relationship between regret and coverage: the concrete bound we get here does not improve on MVP for threshold calibrated coverage. In contrast, our GCACI algorithm (which comes from understanding the relationship between FTRL and coverage) -does- give an algorithm with new state of the art guarantees, which is why we focus our concrete algorithmic exposition on GCACI --- but we're happy to clarify/expand on this in the revision.
> > >
> > > Please let us know if you have any other questions!

---

### Official Review · Reviewer_fRME · 2025-03-12

**Overall Recommendation:** 3

**Summary:**

- This paper studies the relationship between no-regret learning and online conformal prediction
- They show that no-external regret guarantees imply non-trivial coverage guarantees if data is selected stochastically. Moreover, this is *not* true if the data stream is adversarially generated.
- Moving beyond external regret, the authors also consider group-wise regret. Here, they show that even if the data stream is chosen stochastically, no group-wise regret guarantees does *not* imply non-trivial groupwise coverage bounds.
- The authors turn to swap regret. Here, they show that thresholded calibrated coverage is equivalent to no-swap regret. They extend this result to show a similar equivalence between no group-conditional swap regret and multivalid coverage.
- Finally, the authors study the coverage guarantees of online learning algorithms in the FTRL family. Their main theorem in this section relates the miscoverage rate for FTRL with a generic regularizer as a function of the magnitude of the last iterate in FTRL and the gradient of the regularizer. They then instantiate this theorem with specific algorithms in the FTRL family. like gradient descent, to get concrete upper bounds on the miscoverage rates in terms of $T$ and the number of groups $k$. The authors complement their theoretical findings with experiments.

## Update after rebuttal
I thank the authors for their response. As they have satisfactorily addressed my questions and concerns, I will maintain my positive score for this paper.

**Claims And Evidence:**

Yes,  the claims made in the submission supported by clear and convincing evidence.

**Essential References Not Discussed:**

To my best knowledge, the authors covered all related works that are essential to understanding the key contributions of the paper.

**Experimental Designs Or Analyses:**

Yes, I reviewed the experiments in Section 6, but found no glaring issues.

**Methods And Evaluation Criteria:**

Yes, the proposed methods and/or evaluation criteria (e.g., benchmark datasets) make sense for the problem.

**Other Comments Or Suggestions:**

- In the second column of line 26, I think it should be $f(x, y)$ not $f(x)$
- I think it should be "multi valid coverage" instead of "threshold-calibrated coverage" in Definition 2.5
- I think it should be $\tau_t$ not $\Phi_t$ in the Equation in the second column on line 201-202

**Other Strengths And Weaknesses:**

**Strengths**:
- The paper is well-written and easy to follow
- I found the connection between swap regret and thresholded-calibrated coverage interesting
- I found the experimental results to be strong

**Weaknesses**
- Lack of Clarity. In many places, the authors consider transcripts that are random variables themselves (e.g. Theorem 3.2). Unfortunately, I feel that the authors do not do a good job at properly dealing the randomness here. In addition, the authors consider losses throughout the paper, yet the theorem statements are written in terms of gains (i.e. negative losses). More confusingly, in Definition 2.6, regret is defined "with respect to the loss function l", yet the order of the terms in the definition of regret is flipped -- instead of regret being defined as the algorithm's loss minus the competitor's loss, its defined as the competitor's loss minus the algorithm's loss.
- Lack of proper definitions. In many places, the authors do not property define certain objects. For example, what is $\Phi_t$ in the second column of line 202? What is $\mathcal{D}_{\tau}$ is Theorem 3.6?
- Unclear Motivation. I'm not fully convinced of why we should care about coverage guarantees

**Questions For Authors:**

(1) In Theorem 3.2, the transcript $\Pi_T$ is a random variable as the examples $(x_t, y_t)$ are drawn iid. What does it mean to fix a transcript when it's random? Do you mean to fix a realization of the transcript? What does it mean for $\Pi_t$ to have external regret $\gamma$? Is this in expectation or point wise? When stating the coverage guarantees, what exactly is the probability over? Is this theorem essentially saying that if there is algorithm which can obtain *expected* regret $\gamma$ against the pin ball loss, then, with probability at least blah it achieves the stated coverage guarantees?

**Relation To Broader Scientific Literature:**

This paper studies the relationship between various notions of regret minimization and online conformal prediction. In particular, while previous works have established coverage bounds for online conformal prediction under a worst-case adversary, these bounds are not in terms of regret bounds. This work fills this gap by trying to understand what sort of regret bounds can yield bounds on converge for online conformal prediction and vice versa. In addition, while previous works have given OGD-based algorithms for online conformal prediction, this paper extends this to a strictly larger family of FTRL-based online learning algorithms that are able to obtain *both* marginal and multi group coverage guarantees.

**Theoretical Claims:**

As no proofs were included in the main text, I did not verify the correctness of any proofs.

---

> ### Author Rebuttal · Authors · 2025-03-31
>
> Thank you for your feedback! We will fix the typos you’ve pointed out, and address your other specific concerns / questions below:
>
> 1. **$\Pi_T$ as a random variable:** We apologize for any confusion here - In Theorem 3.2, $\Pi_T$ is a realization of the transcript where $(x_t, y_t)$ are drawn from a fixed distribution. This theorem connects the empirical regret $\gamma$ to the empirical coverage, but at a high level uses the fact that expected regret being low implies expected coverage is close to the desired rate. Since we are in a stochastic setting, the empirical and expected values are close to each other with high probability, which is why the result is a high-probability guarantee.
>
> ####
>
> 2. **Definition of $\Phi$-regret:** Thank you for pointing this out - the definition of regret should be the algorithm’s loss minus the competitor’s loss, letting us consider regret with respect to the pinball loss directly. We will update this in revision.
>
> ####
>
> 3. **Undefined quantities:** $\Phi_t$ in line 202 is a typo - this should be $\Pi_t$ instead. $\mathcal{D}_\tau$ in Theorem 3.6 has the same definition as given in Theorem 3.5 - it is the empirical distribution over threshold values defined by the subsequence of the transcript for which the algorithm predicted $\tau$.
>
> ####
>
> 4. **Why coverage guarantees matter:** Achieving a marginal $(1-\alpha)$ coverage guarantee for threshold predictions is equivalent to producing prediction sets that include the true label marginally $(1-\alpha)$ fraction of the time (since the predicted threshold maps to a prediction set through the scoring function); this link similarly applies to groupwise and in general conditional coverage guarantees as well.
>
>     Prediction sets with high coverage guarantees can be used in and of themselves as an interpretation of the uncertainty of a model’s point predictions. More practically, they can also be used to guide decisions for downstream agents. For example, if one is using a learning model in a safety-critical environment, it may be useful to utilize prediction sets with a very high guarantee of coverage to decide which actions minimize or eliminate the possibility of harmful outcomes. [1] formalizes this intuition and shows why prediction sets with coverage guarantees are the right form of uncertainty quantification for risk-averse decision-makers.
>
>  ####
>
> [1]  Kiyani, Shayan, et al. “Decision Theoretic Foundations for Conformal Prediction: Optimal Uncertainty Quantification for Risk-Averse Agents.” ArXiv.org, 2025, arxiv.org/abs/2502.02561.

---

### Official Review · Reviewer_obBc · 2025-03-13

**Overall Recommendation:** 2

**Summary:**

This paper explores the connection between online conformal prediction, which aims to construct prediction sets that cover the true labels with a specified probability, and online learning algorithms minimizing the pinball loss, which aim to achieve a certain regret guarantee. The authors show that standard external regret guarantees ensure marginal coverage when the data is i.i.d.; however, they do not provide meaningful coverage guarantees in adversarial settings or multi-group settings. In contrast, when the data distribution is smooth, the stronger notion of swap regret guarantee is "equivalent" to threshold-calibrated coverage, meaning an upper bound on one directly implies a corresponding upper bound on the other. Finally, the authors extend the adaptive conformal inference algorithm of Gibbs \& Candes (2021) to the multi-group setting and demonstrate that the FTRL algorithm ensures group-wise coverage.

**Claims And Evidence:**

Most of the claims are well supported by counterexamples or theorems. However, I have some questions regarding the equivalence result between swap regret and the threshold-calibrated coverage.
- In Theorem 3.5, the authors provide an upper bound on threshold-calibrated coverage in terms of swap regret. However, the right-hand side approaches zero only if $\rho \rightarrow 0$ and $T_{G,\tau}^3$ grows faster than the swap regret, which is not immediately clear. Consequently, it does not directly follow that any algorithm achieving no swap regret will necessarily lead to the desired threshold-calibrated coverage.
- Ideally, a stronger quantitative equivalence result would establish that a transcript achieves a certain coverage error at most $\gamma$ __if and only if__ it attains a corresponding swap regret bound $R$. However, Theorems 3.5 and 3.6 seem to provide only a qualitative connection rather than a strict equivalence.

**Essential References Not Discussed:**

Some prior works have also explored the use of online learning algorithms for online conformal prediction but with different notions of regret, such as strongly adaptive regret (Bhatnagar et al., 2023) and discounted regret. It would be helpful to discuss these papers to better position the current work within the broader literature.


Bhatnagar, A., Wang, H., Xiong, C., & Bai, Y. (2023). Improved online conformal prediction via strongly adaptive online learning. ICML 2023.

Zhang, Z., Bombara, D. & Yang, H. (2024). Discounted Adaptive Online Learning: Towards Better Regularization. ICML 2024.

**Experimental Designs Or Analyses:**

The experimental design follows that of Bastani et al. (2022) and seems reasonable to me. However, as also discussed in Bastani et al. (2022), for the group-conditioned coverage setting, it may be more meaningful to consider the threshold-calibrated version as the evaluation metric.

**Methods And Evaluation Criteria:**

The first part of the paper focuses on the relationship between regret and coverage, so evaluation criteria are not applicable. In the second part, the authors propose a group conditional ACI algorithm, which is lightweight and appears reasonable for the intended application.

**Other Comments Or Suggestions:**

- In the abstract (Line 027), "follow the perturbed leader" should be corrected to "follow the regularized leader".
- It would be more natural to define the $\Phi$-regret as the opposite of the version currently used in this paper. This way, in Theorems 3.2, 3.5, 3.6, 3.7, 3.8, the regret will be expressed in terms of the pinball loss rather than the negative of pinball loss, which aligns better with standard conventions.
- In Theorems 3.2 and 3.5, the distribution should be $(\alpha,\rho,r)$-smooth instead of $(\rho,r)$-smooth. Also, this definition appears to be related to the definition given in Definition 3.1 of Bastani et al. (2022), which is worth discussing.

**Other Strengths And Weaknesses:**

My main concern is that the paper's main results feel somewhat incomplete.
- In the first part, after establishing the connection between swap regret and threshold-calibrated coverage, a natural follow-up question is whether this insight can lead to a new algorithm for online conformal prediction and how its coverage guarantee compares to the state-of-the-art methods, such as MVP in Bastani et al. (2022). Addressing this would help demonstrate the strength of the equivalence result.
- In the second part, my understanding is that a key issue with the ACI algorithm is that the optimal learning rate $\eta$ depends on unknown quantities of the sequence, as noted in Gibbs & Candès (2022). However, this issue does not appear to be discussed in the current submission.

**Questions For Authors:**

- In the proof of Lemma A.1, the authors state that by the smoothness condition on $\mathcal{D}$, we have $|N_1 - qT| \leq \rho T/2$ (Line 644). Could you please elaborate on this? It seems that the bound should be  $|N_1-  \lceil qT \rceil| \leq \rho T$, and it is unclear where the additional factor of $1/2$ comes from.
- Also, on Line 654, you mention that the final inequality follows from evaluating the expression at the optimal values of $N_1 = qT$ and $N_2 + N_3 = (1-q)T$. Could you clarify in what sense these values are "optimal"?

**Relation To Broader Scientific Literature:**

This papers focus on sequential online conformal prediction problems, initially studied by Gibbs \& Candès (2021) and Gupta et al. (2022). The first part establishes an interesting connection to swap regret, a concept studied in the online learning literature. In the second part, the authors propose and analyze a group-conditional ACI algorithm, which can be viewed as a natural extension of ACI in Gibbs & Candès (2021).

**Theoretical Claims:**

To the best of my knowledge, the proof appears to be correct. However, there are some steps in the derivation that I do not immediately follow, which I have detailed in the "Questions" section.

---

> ### Author Rebuttal · Authors · 2025-04-01
>
> Thank you for your thoughtful and detailed comments! First we would like to note a small change to the statement of Theorem 3.5. It should read: $|Cov(\Pi_T, G_\tau) - q| \leq \frac{\rho}{2} + \frac{\rho}{rn} + \sqrt{\frac{2\gamma}{T_{G, \tau}\alpha r} + \frac{\rho}{\alpha}\left(\frac{1}{r} + \frac{2}{n}\right)}$.
>
> * The cube in $T_{G, \tau}$ was a typo. $\sqrt{\frac{2\gamma}{ T_\tau \alpha r}}$ on line 705 should be $\sqrt{\frac{2\gamma T_\tau}{\alpha r}}$.
> * The result in Lemma A.1. should read $|a-b| \leq \sqrt{\frac{2\gamma}{T\alpha r} + \frac{\rho}{\alpha}\left(\frac{1}{r} + \frac{2}{n}\right)}$, translating to an additional constant term in Theorem 3.5. We explain this below.
>
> ####
> Addressing your questions / comments:
>
> 1. **Dependence on $\rho$ and $T_{G, \tau}$ in Theorem 3.5:** It’s true that $T_{G, \tau}$ may grow slower than $\gamma$ for some $\tau$. However, we think of overall threshold-calibrated coverage as the sum of coverages $Cov(\Pi_T, G_\tau)$, weighted by the size of $G_\tau$. For each threshold $\tau$, either $T_{G, \tau}$ grows slower than $T$, and its weighted contribution $T_{G, \tau}/T$ goes to zero, or it grows on the order of $T$, and Theorem 3.5 tells us that miscoverage goes to zero - so overall threshold-calibrated coverage should converge. We will add this detail in revision.
>     The dependence on $\rho$ is unavoidable, since exact coverage $q$ comes at the $q$-th quantile of the empirical distribution over thresholds. If a single value is allowed to carry $\rho$ of the total probability weight, then the closest we may be able to get is $q \pm \rho/2$.
>
> ####
> 2. **Stronger quantitative equivalence result:** The relationship between swap regret wrt squared loss and mean calibration is not in general tight because of choices in how calibration is normally measured (Linfty/L1 norm vs L2 norm). Similarly we don't believe the relationship between quantile calibration and swap regret wrt pinball loss is tight in the metric we give, but there may be another in which it is. We can note this in the revision.
>
> ####
> 3. **Using threshold-calibrated coverage as a metric:** We agree that it would be interesting to compare the performance of existing swap regret algorithms against MVP using threshold-calibrated coverage as a metric.
>
>     However, one of the key points demonstrated in this paper is that achieving threshold-calibrated coverage is at least as difficult as achieving no swap-regret, which requires more computational overhead than obtaining external regret (no-swap-regret algorithms are based on finding fixed points of various sorts). Thus, there is a necessary tradeoff in terms of algorithmic simplicity if threshold-calibrated coverage is what we want - and we believe that MVP already occupies a good place in the tradeoff space, as a more complicated algorithm with a stronger qualitative bound. But as we see in Sections 4 & 5 (and in experiments), if all we want is group conditional coverage (giving up on threshold calibration), then MVP is overkill - we can get much simpler algorithms with faster convergence to the desired coverage rate. For marginal coverage, ACI already demonstrated this was possible compared to MVP - it is a strikingly simple and efficient algorithm that obtains marginal (not threshold calibrated) coverage. In our work we show that we can recover the same kind of simple, efficient algorithms even with group conditional coverage.
>
> ####
> 4. **How an optimal learning rate $\eta$ is chosen:** In both our paper and in ACI, choosing $\eta = 1$ gives the best rate of convergence towards desired coverage ($O(1/\sqrt{T}$ for ours, $O(1/T)$ for ACI). The discussion of optimality of $\eta$ in Gibbs & Candes (2021) is to do with balancing between having quick rates of convergence and having low volatility in predicted thresholds. The larger $\eta$ is, the quicker coverage converges and the more adaptive predictions are to adversarial shifts. The smaller $\eta$ is, the less predictions fluctuate round to round. They use this to describe how $\eta$ may be chosen in conditions where the distribution shift can be measured. Our work inherits the same balance between convergence and volatility, but since we are interested specifically in adversarial coverage we didn’t discuss this in detail. Empirically, we found indeed that the convergence rate improves the closer to 1 that $\eta$ gets.
>
> ####
> 5. **Lemma A.1 questions:** We should have the bounds $|N_1 - qT|, |N_2 + N_3 - (1-q)T| \leq \rho T/2 + \rho r T/ n$. The bound of $\rho T / 2$ is true if $a$ is the exact $q$-th quantile (or the closest value to it). Since $a$ may be at a distance of $1/n$ away from this value, there should be an additional term of $\rho r T / n$.
>
>     The meaning of “optimal” was in reference to getting a lower-bound on the terms involving $N_1, N_2$ and $N_3$. Based on the updated bounds above, this appears as an additional term $-(b-a)(\frac{\rho T}{2} + \frac{\rho rT}{n})$ on Line 652.

---

> > ### Comment · Reviewer_obBc · 2025-04-06
> >
> > I thank the authors for their responses and for addressing my concerns. I have the following remarks:
> > - Regarding point 2, I am not sure I fully understand the authors' comment. First, it seems that the swap regret is with respect to the pinball loss rather than the squared loss. Moreover, regarding the discussions on $\ell_{\infty}/\ell_{1}$-norm v.s. $\ell_2$-norm, I think the authors are suggesting that the threshold-calibrated coverage in Definition 2.4 measures calibration in terms of the $\ell_{\infty}$-norm, since $\gamma$ is the maximum of the coverage error among all the groups. Accordingly, the authors propose that we may establish a more quantitative equivalence by choosing a different norm for measuring coverage error. Could the authors please clarify?
> > - Regarding point 3, the authors appear to suggest that the first part of their paper should be viewed as a *negative result*, showing that "achieving threshold-calibrated coverage is at least as difficult as achieving no swap-regret". However, this was not my initial impression when reading the paper. On the contrary, the authors state that "This [the equivalence result] gives new algorithms for guaranteeing group conditional multivalid coverage", which seems to frame the result more positively. If the intent is indeed to convey a negative result, it would be helpful for the authors to make this message more explicit. Furthermore, while it is true that achieving sublinear swap regret is generally more computationally demanding than external regret, in the context of online conformal prediction, we are dealing with a one-dimensional pinball loss, which may be more tractable. Thus, I am also uncertain about the strength of this negative result.
> > - Regarding Lemma A.1, I may be missing something obvious: if $a$ is the exact $q$-th quantile of the set $\\{\tau_i\\}\_{i=1}^T$, then $a$ corresponds to the $\lceil q T\rceil$-th smallest element in $\\{\tau_i\\}\_{i=1}^T$. Given the smoothness assumption on $\mathcal{D}$, there can be at most $\rho T$ elements in the set equal to $a$. This suggests that $|N_1 - \lceil q T \rceil | \leq \rho T$, rather than $\rho T/2$. In any case, this appears to be a minor issue.

---

> > > ### Author Response · Authors · 2025-04-07
> > >
> > > Thank you for engaging --- we know reviewing is is important/demanding work and we sincerely appreciate it. In response to your questions:
> > >
> > > 1. Yes, in our paper we give a qualitative (i.e. non-rate preserving) equivalence between threshold calibrated coverage on the one hand and swap regret with respect to pinball loss on the other. We bring up (mean) calibration and swap regret with respect to squared error only to point out that in the more standard setting in which there is an equivalence between a kind of "calibration" and swap regret, the situation is analogous --- the equivalence is qualitative, not tight/rate preserving. In the case of mean calibration, there are multiple competing measures of calibration error, and so it is not surprising that they do not have a tight quantitative relationship with a fixed measure of regret (they differ amongst themselves). We here suggest that the situation may be similar in the case of threshold calibrated coverage. But we don't want to belabor the analogy --- our only point is that you are correct that the relationship is not rate preserving, but that this is not unexpected.
> > >
> > > ####
> > >
> > >
> > > 2. The relationship between swap regret and threshold calibrated coverage is not a negative result per-se --- swap regret is a stronger condition than external regret, and threshold calibrated coverage is a stronger condition than coverage on its own. There is an algorithm for obtaining swap regret in one-dimensional regression problems with respect to one dimensional predictions just as we use here --- see e.g. "Oracle Efficient Online Multicalibration and Omniprediction" from SODA 2024 --- and this algorithm is very similar to more standard swap regret algorithms like Blum/Mansour. In particular it requires computing the top eigenvector of a matrix at every iteration, and so is not as lightweight as simple algorithms like online gradient descent. So, our equivalence indeed gives a new algorithm for obtaining threshold calibrated coverage, but not one that would be substantially more lightweight than the existing algorithm of Bastani et al (we explicate this in more detail in our response to Reviewer b1E4's most recent questions). This is why we focus our empirical evaluation on our online gradient descent based algorithm, which is indeed substantially more practical (in terms of both convergence rates and per-iterate run-time) than any previously known algorithm for obtaining group conditional coverage in the online adversarial setting.
> > >
> > > ####
> > >
> > > 3. The confusion might arise from how we are defining $q$-th quantile. Since we are trying to minimize pinball loss, we are looking for the value that comes closest to covering q of the probability weight (or equivalently $qT$ of the values in the set defining the distribution), which is not the same thing as the smallest value that covers *at least* q of the probability weight (since this isn't the standard definition, we will clarify this in revision). Since the point $a$ along [0,1] where we go from covering less than $qT$ of the values to more can itself hold at most $\rho$ of the probability weight, the worst-case scenario is when below $a$ there is (approximately) $q - \rho/2$ and above there is $(1-q)  + \rho/2$ of the probability weight.
> > >
> > > Thanks again for engaging, and please let us know if there are any additional questions!

---

### Official Review · Reviewer_8yW2 · 2025-03-24

**Overall Recommendation:** 3

**Summary:**

This paper explores how to make better predictions with reliable uncertainty estimates in challenging, real-world settings—like when data changes over time or comes from many different groups. It focuses on *conformal prediction*, a method that builds a set of likely outcomes for each input, with a guarantee that the true answer falls inside that set a desired percentage of the time (say, 90%). Traditionally, conformal prediction works well in nice, stable environments, but struggles in harder, online or adversarial scenarios. The key insight of this paper is that existing learning techniques based on *external regret* (which just try to do as well as the best fixed action in hindsight) aren't enough to ensure these prediction sets stay reliable—especially when we care about fairness or accuracy across different groups. Instead, the paper shows that a stronger concept called *swap regret*—which says “I wouldn’t have done better even if I had consistently changed certain decisions in a smarter way”—is exactly what's needed to ensure these coverage guarantees hold, even in complex settings. [That's my hand wavy understanding]
Using this idea, the authors design a simple and effective new algorithm called GCACI that can adjust its uncertainty sets over time for many groups simultaneously. They prove this method works in theory, and show in experiments that it beats previous approaches by achieving better coverage, more quickly, and with less computational effort.

**Claims And Evidence:**

The core claims are mostly theoretical and comes with formal proofs. The paper essentially provide this:

- External regret is not sufficient for coverage guarantees: counter-example appears in adversarial setting or even iid but with group structure. Instead Swap regret is both necessary and sufficient for coverage.

- Derive new conformal algorithms from no-swap-regret learning and achieve group-conditional coverage

**Essential References Not Discussed:**

Adaptive Conformal Inference by Betting by (Aleksandr Podkopaev, Darren Xu, Kuang-Chih Lee)

**Experimental Designs Or Analyses:**

The experiments section can be improved with richer experiments

**Methods And Evaluation Criteria:**

The evaluations are quite weak and more benchmark could be done with more datasets

**Other Comments Or Suggestions:**

NA

**Other Strengths And Weaknesses:**

The paper is reasonably well written and the core contributions are solid.

The weakness is mostly the experimental section.

**Questions For Authors:**

- Your coverage guarantees rely on an smoothness condition on the empirical distribution of true thresholds . This assumption is key for bounding the difference between pinball loss regret and coverage error. Could you clarify how realistic this assumption is in practice, especially in adversarial or highly structured environments where the distribution of may be heavily clustered or even discrete?
Have you observed situations (empirically or theoretically) where violation of this smoothness significantly affects coverage, and do you foresee ways to either relax the assumption or detect when it may not hold?

- In your setup, the adversary is allowed to choose the sequence of examples (or losses), but the smoothness assumption effectively constrains the sequence of threshold values. How should we interpret the adversary’s power in this setting? Is the smoothness constraint best understood as an assumption on the data-generating process (i.e., nature, not an adversary), or are there adversaries within your model who can adaptively shape  while still satisfying smoothness?
More generally, could you comment on the interaction between adversarial flexibility and the structural assumptions needed for regret-to-coverage equivalence?


 "## update after rebuttal"

The authors provided some more clarifications about the novel contributions, which is nice; However, several points regarding the importance of swap regret and strength of the propositions compared to previous (group) conditional coverage are quite unclear to me. I think the paper would benefit from more rewritting and concret examples. That being said I am globally positive but cannot increase my score to full accept

**Relation To Broader Scientific Literature:**

The paper fall into connecting conformal prediction and online learning. Mostly borrow ideas from the latter to improve understanding of the former.

**Theoretical Claims:**

Just skim quickly and did not check in details.

---

> ### Author Rebuttal · Authors · 2025-03-31
>
> Thank you for your detailed feedback! We will try to address your concerns and comments.
>
> First, as a clarification: there are two kinds of relationships we develop in this paper - (1) the relationship between the *properties* of no-regret and of conformal coverage guarantees, agnostic to the algorithm used. (2) specific algorithms that are able to achieve both kinds of properties, even if the properties do not imply one another. One of the motivations for this paper was identifying that though external regret and coverage guarantees do not imply each other (i.e. an arbitrary algorithm that achieves no external regret cannot guarantee coverage guarantees, and vice versa), there exist specific algorithms that are able to achieve both properties. GCACI is an example of this - the analysis for coverage does not go through regret.
>
> 1. **Smoothness condition on distribution of thresholds:** Smoothness is a mild condition and we can view it either as an assumption on the adversary (a “smoothed analysis” like assumption --- for example, the adversary can pick an arbitrary score which is then perturbed by small amounts of noise), or alternately as a condition that we can enforce ourselves. If we want to enforce it ourselves we simply perturb the scores  from a uniform distribution $[-\epsilon, \epsilon]$ before passing them to the algorithm. We then randomize the algorithm’s threshold in this range. This doesn’t require us to make assumptions on the adversary at all.
>
>     Note that it is not possible to draw a direct relationship between coverage and regret without some kind of smoothness guarantee. As you mention, the relationship between regret wrt pinball loss and coverage relies on the non-conformity scores being sufficiently spread out. If, for example, the distribution over scores is a point mass on some value $a$, the only achievable threshold-calibrated coverage rates are 0 and 1 - to achieve low miscoverage for any quantile $q$ in between, some assumption of smoothness is necessary. Similarly, in the stochastic setting, marginal $(1-\alpha)$-coverage could be achieved with predicted values arbitrarily far away from $a$ as long as $(1-\alpha)$ fraction of the values were below $a$, but this obviously doesn’t translate into a regret guarantee.
>
>     We also emphasize that the smoothness guarantee is only required for algorithms achieving coverage through swap regret. Our new algorithm GCACI achieves coverage through properties of the class of FTRL algorithms, and thus no assumptions on smoothness need be made. This is an advantage over the MVP algorithm we compare against, which does require it.
>
> ####
>
> 2. **Power of adversary when smoothness is required:** Since smoothness can be enforced posthoc via the perturbation strategy, we do not typically enforce any kind of constraints on the adversary directly.
>
> ####
>
> 3.  **Experiments not rich enough:** We included experiments to compare against all the benchmarks described in Bastani et al (2022), which to our knowledge details the only other algorithm for groupwise coverage in adversarial settings. We would be happy to include additional experiments that may be relevant to evaluation, if the reviewer has any particular benchmarks in mind.

---

> > ### Comment · Reviewer_8yW2 · 2025-04-04
> >
> > Thanks for the comments and clarification.
> >
> > > We included experiments to compare against all the benchmarks described in Bastani et al (2022), which to our knowledge details the only other algorithm for groupwise coverage in adversarial settings
> >
> > I am not sure to understand this comment. In my understanding the Line of work by Gibbs & Candes, with the online learning View can adapt to adversarial settings. Plus on a other follow up they provided an extensive analysis on conditional coverage. Mainly the idea is that conditional coverage can we written as an orthogonality condition wrt all measurable function of the test point. Thus to relax this notion they provided some restricted class of function that covers conditional group coverage.
> >
> > As such, most of the online CP follow up could be combined with these type of restrictions. So I am quite confuse by this comment.
> >
> > Would be Nice if the authors can clarify. Sorry if I am missing some obvious point..

---

> > > ### Author Response · Authors · 2025-04-04
> > >
> > > Hi! Thanks for engaging, we really appreciate it. There are two lines of work that are relevant to this discussion, so lets try and disentangle them.
> > >
> > > There is the line of work on conformal prediction in online adversarial settings, using 1-dimensional gradient descent on pinball loss and variants. This line started with "Adaptive Conformal Inference Under Distribution Shift" by Gibbs and Candes. These algorithms update a single parameter (mapping to the threshold on the conformity score) and the coverage analysis depends on the fact that when one runs gradient descent on the pinball loss of a 1 dimensional quantile estimate, the iterates have magnitude bounded by 1. This line of work does not give groupwise coverage guarantees (and cannot, as it uses a 1 dimensional parameterization of the threshold).
> > >
> > > There is also a line of work on conditional coverage in conformal prediction. We are guessing the paper you are referring to here is "Conformal prediction with conditional guarantees" by Gibbs, Cherian, and Candes. The conditional guarantees in this work (as in the prior work in this literature) follow from the first order optimality conditions of pinball loss when optimizing a d-dimensional linear function mapping features to thresholds. In the groupwise case, the d dimensions correspond to indicator functions for d groups. When you run gradient descent on this parameterization, the iterates are no longer bounded by 1 --- the ACI analysis breaks. It is possible to analyze this algorithm --- this is one of the contributions of our paper --- but it requires a new analysis and gets a fundamentally different kind of bound. This is new to our work and hasn't been done previously. As we note in our submission, there is an independent and concurrent paper that discovers the same algorithm ("Gradient Equilibrium in Online Learning: Theory and Applications" by Anastasios  Angelopoulos, Michael  Jordan, Ryan Tibshirani) --- so we are confident that this work is not implicit in the prior literature.
> > >
> > > We're happy to engage further if you have any other questions --- as we said, we appreciate the opportunity to interact and are grateful for the time and effort you have spent in reviewing!

---

### Decision · Program_Chairs · 2025-05-01

**Decision:**

Accept (poster)

**Comment:**

This work identifies a novel connection between coverage conditions in conformal prediction and classes of no-regret algorithms in online learning. Specifically, group (multivalid) coverage conditions are identified as linked to swap regret, for which follow the perturbed/regularized leader achieve no regret. Theoretical and experimental implications of these connections are illuminated, which the reviewers unanimously assess to be meritorious for the machine learning research community. While there is some disagreement on the novelty of the proofs and discussion of related works, this is not enough to be disqualifying. Moreover, the paper is well-written and contributes fundamentally new results to an important area of machine learning.